

# Deep Circulation in the South China Sea Simulated in a Regional Model

Xiaolong Zhao[1,2], Chun Zhou[2], Xiaobiao Xu[3], Ruijie Ye[2], Jiwei Tian[2] and Wei Zhao[2]

[1]North China Sea Marine Forecasting Center, State Oceanic Administration, Qingdao, 266061, P. R. China.
[2]Key Laboratory of Physical Oceanography/CIMST, Ocean University of China and Qingdao National Laboratory for Marine Science and Technology, Qingdao 266100, P. R. China.
[3]Center for Ocean-Atmospheric Prediction Studies (COAPS), Florida State University, Tallahassee, FL, USA.

*Correspondence to:* Wei Zhao (weizhao@ouc.edu.cn)

**Abstract.** The South China Sea (SCS) is the largest marginal sea in the northwest Pacific Ocean. In this study, deep circulation in the SCS is investigated using results from eddy-resolving, regional simulations using the Hybrid Coordinate Ocean Model (HYCOM) verified by continuous current-meter observations. Analysis of these results provides a detailed spatial structure and temporal variability of the deep circulation in the SCS. The major features of the SCS deep circulation are a basin-scale cyclonic gyre and a concentrated deep western boundary current (DWBC). Transport of the DWBC is ~2 Sv at 16.5 °N with a width of ~53 km. Flowing southwestward, the narrow DWBC becomes weaker with a wider range. The model results reveal the existence of 80- to 120-day oscillation in the deep northeastern circulation and the DWBC, which are also the areas with elevated eddy kinetic energy. This intraseasonal oscillation propagates northwestward with a velocity amplitude of ~1.0 to 1.5 cm s$^{-1}$. The distribution of mixing parameters in the deep SCS plays a role in both spatial structure and volume transport of the deep circulation. Compared with the northern shelf of the SCS with the Luzon Strait, deep circulation in the SCS is more sensitive to the large vertical mixing parameters of the Zhongsha Island Chain area.

## 1. Introduction

The South China Sea (SCS, Fig. 1) is the largest marginal sea in the Southeast Asian Waters, with an area of approximately $3.5 \times 10^6$ km$^2$ and a depth exceeding 4000 m in the central basin (Wyrtki, 1961). It is connected to the surrounding waters mostly by shallow straits: The Taiwan Strait to the East China Sea in the north, the Karimata Strait to the Java Sea in the south, and the Mindoro Strait to the Sulu Sea in the southeast. The 355 km-wide Luzon Strait, with a sill depth of ~2400 m, is the only deep connection between the SCS and its ambient oceans. There, cold and salty (thus dense) North Pacific Deep Water (NPDW, with potential temperature and salinity of ~1.79 °C and 34.64 psu; Mantyla, 1975; Zhao et al., 2016) penetrates the SCS basin through the deepwater overflow in the Luzon Strait driven by the baroclinic pressure gradient between the Pacific Ocean and the SCS (Qu et al., 2006a; Zhao et al., 2014; Zhou et al., 2014, 2018). Since the SCS is closed below 2400 m, the incoming NPDW eventually upwells as a result of enhanced mixing (~$10^{-3}$ m$^2$ s$^{-1}$; Tian et al., 2009; Alford et al., 2011; Yang et al., 2016) and



exits the SCS either in the intermediate layer through the Luzon Strait back to the Pacific Ocean (Chao et al., 1996;
Chen and Huang, 1996; Li and Qu, 2006; Qu et al., 2000; Tian et al., 2006; Zhang et al., 2015; Gan et al., 2016) or
in the upper layer through several shallow straits in the southern part of the SCS to the Java and Sulu Seas (e.g., Qu
et al., 2009; Yaremchuk et al., 2009). This three-dimensional circulation constitutes the SCS throughflow (Qu et al.,
2006b), serving as a heat and freshwater conveyor that is climatologically important on regional and global scales
(e.g., Gordon et al., 2012).

As a key element of the SCS circulation, the deepwater overflow through the Luzon Strait has been observed in a

number of studies (e.g., Wang, 1986; Liu and Liu, 1988; Qu et al., 2006a; Song, 2006; Tian et al., 2006; Chang et al.,
2010; Yang et al., 2010, 2011; Tian and Qu, 2012; Zhao et al., 2014; Zhou et al., 2014; Zhao et al., 2016; Ye et al.,
2019), and its volume transport as well as the water properties are now relatively well defined. Based on data from
moorings deployed at two locations for 3.5 years, Zhou et al. (2014) estimated a mean transport of 0.83 Sv (1 Sv =
$10^6$ m$^3$ s$^{-1}$) in the Bashi Channel and 0.88 Sv further downstream in the Luzon Trough (which includes some
additional minor contribution through the Taltung Canyon north of the Bashi Channel). More recently, Zhao et al.
(2016) used results from ten current meters at three mooring locations in the Bashi Channel and estimated a similar
eight-month mean transport of 0.78 Sv (with a total rms error of 0.18 Sv). The overflow transport exhibits a
significant seasonal variability (with a higher transport in October-December and a lower transport in March-May),
corresponding well with the seasonal variation of the density difference between the SCS and the Pacific Ocean
close to the sill depth (Zhou et al., 2014) and a significant intraseasonal variability on a near 30-day timescale,
which is close to the resonance period of the deep channel in the Luzon Strait (Zhao et al., 2016). The time series
from 2009 to 2013 indicates an interannual variability, but longer observations are needed to determine long-term
variability.

Compared to the deepwater overflow in the Luzon Strait, much less is known about the deep circulation in the

SCS. In general, a cyclonic circulation with an intensified deep western boundary current (DWBC) is expected,
following the classical Stommel-Arons abyssal circulation theory (Stommel and Arons, 1960a, 1960b). The
temperature, salinity, and tracer distributions of the World Ocean Database 2001 indicate such a cyclonic circulation
in the deep SCS (Qu et al., 2006a). A similar basin-scale cyclonic circulation, with an estimated mean transport of
3.0 Sv, is suggested by Wang et al. (2011) based on an analysis of the ocean climatology database, the Generalized
Digital Environment Model (GDEM; Carnes, 2009). Recently, an array of six current meter moorings was deployed



off the eastern slope of the Zhongsha Islands from August 2012 to January 2014 (Zhou et al., 2017). Results from
these direct measurements show, for the first time, the existence of the DWBC in the deep SCS basin, with a volume
transport of 1.65 Sv and high temporal variability around 90 days. This mooring array in Zhou et al. (2017) is used
in the present study.

Numerical models are also used to study the deep circulation in the SCS. Chao et al. (1996) using a 0.4 ° three-

dimensional, climatology-driven circulation model show a deep cyclonic circulation in the deep SCS but without
clear DWBC. Lan et al. (2013, 2015), based on results of 0.5 ° simulations, suggest that the basin-scale deep
circulation is controlled by the deep overflow from Luzon Strait. In their simulation, a basin-scale cyclonic gyre is
prominent during July-September and hardly identified during January-March. With data assimilation and higher
resolution, Shu et al. (2014) and Xu and Oey (2014) show a complicated three-layer circulation in the SCS, cyclonic
in the upper layer, anticyclonic in the middle, and cyclonic in the deep. With 1/12 ° MITgcm, Wang et al. (2017)
simulated a strong north deep circulation comparable with the DWBC. Earlier simulating studies indeed indicated
the general cyclonic pattern of the deep SCS circulation and the existence of the DWBC. However, numerous
discrepancies exist among different simulation results: First, the accurate location of the DWBC is controversial. For
example, Lan et al. (2013, 2015) simulated deep circulation flows southwestward off the western slope of the
Zhongsha Islands, while Shu et al. (2014) and Xu and Oey (2014) indicated the DWBC flows off the eastern slope
of the Zhongsha Islands. Since the DWBC is due to the Luzon Strait overflow and the β effect, whether the model
horizontal resolution is sufficient to distinguish the deep Luzon Strait (~15 km wide at 2000 m depth, which is the
time mean upper interface of the overflow, see Zhao et al., 2016) could be the reason. Second, in most simulations
there is a strong cyclonic or anticyclonic circulation cell at the southwest part of the deep circulation under week
mixing: a separate cyclonic circulation in Chao et al. (1996) and Shu et al. (2014), while there is an anticyclonic one
in Xu and Oey (2014). Due to the lack of field observations, simulation results of the deep circulation in the SCS
need to be verified before being employed to the discussion of the spatio-temporal characteristics of the deep
circulation in the SCS.

Enhanced mixing is a well-observed feature in the SCS. The observations of Tian et al. (2009) and Alford et al.

(2011) show diapycnal diffusivity in the SCS and the Luzon Strait increases from about $10^{-3}$ $m^2$ $s^{-1}$ at 1000 m to $10^{-2}$
$m^2$ $s^{-1}$ near the sea floor. This is about two orders of magnitude higher than that in the North Pacific Ocean and is
furnished by energetic internal waves induced by the prominent bathymetry in the Luzon Strait (Niwa and Hibiya,



2004; Jan et al., 2007; Tian et al., 2003, 2006). Based on hydrographic measurements with fine scale
parameterizations from 335 stations (477 casts), Yang et al. (2016) recently obtained the three-dimensional
distribution of turbulent mixing in the SCS for the first time. Two mixing "hotspots" were identified in the bottom
waters in the northern shelf of the SCS with the Luzon Strait and the Zhongsha Island Chain areas (their Fig. 4),
largely due to internal tide, bottom bathymetry, and near-inertial energy. Previous studies have shown enhanced
mixing plays a role in deep circulation in both the Pacific Ocean and the Luzon Strait. Furue and Endoh (2005)
indicated the deep Pacific Ocean diffusivity contributes to enhanced production of the Antarctic Bottom Water in the
model. The northward transport of the deep meridional overturning circulation across the equator in the Pacific
Ocean is stronger with the intense mixing than with weak mixing (Endoh and Hibiya, 2006; their Fig. 3). Zhao et al.
(2014) suggested that enhanced mixing in the SCS and the Luzon Strait was the primary driving mechanism for the
deep circulation in the Luzon Strait, since it is a key process responsible for the density difference between the
Pacific Ocean and the SCS. Based on a simulated tidal mixing scheme, Wang et al. (2017) indicated the tide-induced
diapycnal mixing in the Luzon Strait would have a negative effect on driving the cyclonic SCS deep circulation,
although without the feature of two mixing "hotspots". Since the mixing is very strong and unevenly distributed in
the deep SCS, it is necessary to modify the mixing scheme in the ocean model to be consistent with observed three-
dimensional distribution of mixing. Nevertheless, previous numerical studies simulated the deep circulation with
homogeneous or simulated vertical mixing parameters in the deep SCS, and one wonders about the sensitivity of the
SCS deep circulation to the observed distribution of mixing.

Given the lack of observations and inadequate quality control, detailed structures of circulation in the deep SCS

have not been mapped out and described adequately. Combining the mooring array in Zhou et al. (2017) with results
from eddy-resolving model simulations, the present study investigates deep circulation under enhanced mixing in
the SCS. The paper is organized as follows. After the introduction, the data and model configuration are described in
Sect. 2. Section 3.1 presents the model results compared with observations. Section 3.2 is devoted to the horizontal
pattern of mean circulation. Variability of deep circulation is discussed in Sect. 3.3, and Sect. 3.4 examines
sensitivity to distribution of mixing. A summary and discussion follows in Sect. 4.
**2. Data and Model Configuration**
As part of the SCS mooring array, an array of six bottom-anchored moorings was deployed off the eastern slope of





the Zhongsha Islands between 28 August 2012 and 11 January 2014 (M1-M6, see Fig. 1 for locations). Twenty-nine
Aanderaa Data Instruments RCM Seaguard current meters were utilized to measure the horizontal current of the
DWBC at nominal depths of 2000 m, 2500 m, 3000 m, 3500 m, and 4000 m, with generally 500 m resolution
vertically. Details pertinent to these moorings are shown in Table 1. All current meters were configured to record
data at a sample interval of one hour. Detailed results are discussed in Zhou et al. (2017). Here, we use the observed
mean velocity section to examine the simulated time mean structure of the DWBC.

The regional simulation is similar to that of Zhao et al. (2014). The general circulation model used was the Hybrid

Coordinate Ocean Model (HYCOM; Bleck, 2002; Chassignet et al., 2003) configured with a horizontal resolution of
$1/12\,°$ (~9 km resolution in our area of interest). The computational domain, which extends from 4 °N to 25 °N and
105 °E to 125 °E (Fig. 1), includes the SCS and part of the northwestern Pacific Ocean. A total of 32 vertical hybrid
layers are configured with density referenced to 2000 m ($\sigma_2$, kg m$^{-3}$): 28.10, 28.90, 29.70, 30.50, 30.95, 31.50, 32.05,

126    32.60, 33.15, 33.70, 34.25, 34.75, 35.15, 35.50, 35.80, 36.04, 36.20, 36.34, 36.46, 36.56, 36.64, 36.70, 36.74, 36.78,

36.82, 36.84, 36.86, 36.88, 36.92, 36.96, 37.01, and 37.06. The bottom topography is from version 13.1 of Smith
and Sandwell (1997) with 1' resolution. The simulation was initialized with rest and January temperature and
salinity fields from the third version of monthly $1/4\,°$ ocean climatology GDEM (Carnes, 2009). Despite the fact that
surface forcing is significant in this region as regulating the upper layer circulation, evidence of surface forcing to
the deep layer dynamics has not yet been found. Since the current work is designed to be a process study, surface
forcing was not applied in the experiments. All lateral boundaries were closed with no normal flow, within a 19-grid
buffer zone near the eastern boundary, the modeled temperature and salinity are restored toward the same (monthly)
climatology with an e-folding time of 0.5-32 days that increased with distance from the boundary. The bottom stress
was parameterized using a quadratic drag law at the lowest 10 m, with a constant drag coefficient $C_D = 2.5 \times 10^{-3}$.

Based on similar configurations with all of the numerical experiments started from rest and integrated for 10 years,

Zhao et al. (2014) studied the deep water circulation in the Luzon Strait, which was in good agreement with the
observations. We modified the K-profile parameterization (KPP; Large et al., 1994) mixing scheme in accordance
with the two observed mixing "hotspots" found in Yang et al. (2016). Thus, the control run was configured with
larger vertical mixing parameters, in which the diapycnal diffusivity beneath 1000 m were set to $10^{-3}$ m$^2$ s$^{-1}$ in both
the north shelf of the SCS with the Luzon Strait (109-122 °E, 18-23 °N) and the Zhongsha Island Chain area (109-
122 °E, 14-17 °N, red boxes in Fig. 1). To examine the impact of mixing, four sensitivity experiments were used with



the same configuration as the control run, but with different mixing schemes: Following Zhao et al. (2014), Exp-5
and Exp-3 were configured with the native KPP scheme as background mixing of $10^{-5}$ m$^2$ s$^{-1}$ and the diapycnal
diffusivity beneath 1000 m in the SCS and the Luzon Strait (west of 122 °E) as $10^{-3}$ m$^2$ s$^{-1}$, respectively. Exp-3A and
Exp-3C were configured with the lager vertical mixing parameters in different areas, in which the diapycnal
diffusivity beneath 1000 m were set to $10^{-3}$ m$^2$ s$^{-1}$ in the north shelf of the SCS with the Luzon Strait (109-122 °E, 18-
23 °N) and the Zhongsha Island Chain area (109-122 °E, 14-17 °N), respectively. In order to obtain a steady state of
the deep circulation in the SCS, we integrated all of the numerical experiments for 20 years and averaged the last
five years as the simulated annual mean results mentioned below (the control run has been stable during the last 10
years).
**3. Key Results**
Observations from six moorings allow us to examine the simulate time mean structure of the DWBC and results
from eddy-resolving model simulations are used to further investigate the structure and mechanisms of the deep
circulation in the SCS.
**3.1 DWBC in the SCS**
Figure 2 presents a comparison between the observed and simulated section view of the mean current in the deep
western boundary of the SCS. Based on Zhou et al. (2017) and considering that the DWBC generally follows the
topography, the observed current is re-coordinated into the cross-section, generally along the isobaths with positive
direction pointing to the southwest. Observations at M5 and M6 are projected to the section (M1-M4). The
simulated time-mean structure of velocity is a zonal section view of 15.4 °N for the control run close to these six
moorings and indicated in the Fig. 1. Consistent with the observations, a bottom intensified current is simulated
flowing southwestward off the eastern slope of the Zhongsha Islands. This is different from Lan et al. (2013, 2015)
but similar with Shu et al. (2014) and Xu and Oey (2014). It appears that a horizontal resolution of 0.5 ° is not
sufficient to resolve the deep Luzon Strait accurately, resulting in an inaccurate position of the DWBC in the
simulation. The DWBC weakens upward, with its upper interface lying at around 2000 m. Horizontally, the model
accurately reproduces the observed main axis of the DWBC (comparable with M1 and M2) and a recirculation
(comparable with M4 an M5). The DWBC is ~100 km wide, with its core leaning on the slope of Zhongsha island.
This modeled and observed DWBC is significantly narrower than Wang et al. (2011). Note that the simulated



DWBC (4 cm s$^{-1}$) and recirculation are stronger than the observations (2 cm s$^{-1}$) since the source, deepwater
overflow in the Luzon Strait, is the same status (1.2 to 0.8 Sv; Zhou et al., 2014; Zhao et al., 2016). As expected, the
control run shows reasonable agreement with the cross-section observations.
**3.2 Mean Circulation Pattern**
To examine the simulated large-scale deep circulation in the SCS, we calculated the mean transports along four
zonal sections (13.5 °N, 15.0 °N, 16.5 °N and 18.0 °N) of each layer including the 25th to 30th from 110 °E to 121 °E
(Fig. 3) for the control run. The cumulated transport of the 27th ($\sigma_2$=36.86 kg m$^{-3}$, ~3000-3500 m) layer shows a
northward current in the southern part of the western boundary (near 114 °E in sections of 13.5 °N and 15.0 °N) that
belongs to the anti-cyclonic middle layer of the SCS circulation (e.g., Gan et al., 2016; Shu et al., 2014; Xu and Oey,
2014), while the 28th ($\sigma_2$=36.88 kg m$^{-3}$, ~3500-4000 m) and 29th ($\sigma_2$=36.92 kg m$^{-3}$, ~4000-4200 m) layers show a
consistent southward DWBC at different latitudes. The mean transport per unit width (in m$^2$ s$^{-1}$) from the 28th layer
shows a strong deep cyclonic circulation in the SCS (Fig. 4a), and the 29th layer mostly presents the deep circulation
in the Luzon Strait (Fig. 4b). Therefore, here we calculate the mean transport per unit width from the 28th to 29th
layer to describe the pathway of deep circulation in the SCS (Fig. 5).

The major features of the SCS deep circulation are a basin-scale cyclonic gyre and a western intensification.

Driven by the baroclinic pressure gradient between the Pacific Ocean and the SCS in the Luzon Strait, deepwater
overflow spills into the SCS mostly through two gaps in the Heng-Chun Ridge (as WG2 and WG3 in Zhao et al.,
2014) along the 3800 m and 4000 m isobaths, respectively. With a confluence off the northern shelf, the current
flows southwestward and then turns southward near 116 °E, 18 °N as an intensified DWBC along the eastern slope of
the Zhongsha Islands. Restricted by the topography, the DWBC divides into two branches at 115 °E, 15.5 °N. A
strong southwestward branch follows the western boundary southwestward and another goes southeastward near M4.
The rest of the DWBC travels to the deep basin in the south and then turns northeastward into the middle basin,
presenting a cyclonic pattern that makes the inflow water spread to nearly the entire SCS deep basin. We cumulated
the mean transports along these four zonal sections from different layers to the 29th in order to quantitatively
describe the deep circulation in the SCS (Fig. 6). The volume transport of the DWBC is ~2.0 Sv at 16.5 °N (from the
27th to 29th layers) with a width of ~53 km, in agreement with the observed transport (1.65 Sv) and larger than the
deepwater overflow in the Luzon Strait (1.2 Sv), which may be related to the entrainment of water from the interior
ocean due to enhanced diapycnal mixing in the northeastern SCS (Tian et al., 2009; Yang et al., 2016). While



flowing southwestward with an upwelling process, the DWBC becomes weaker and gets a wider range: Transport of
the DWBC becomes ~1.2 Sv (from the 28th to 29th layers) with a width of ~140 km at 13.5 ̊N.

### 3.3 Temporal Variability of the Deep Circulation

The model results reveal the existence of energetic intraseasonal variability in the SCS deep circulation. As shown in
Fig. 7a, large eddy kinetic energy (EKE) areas appear in the deep northeastern circulation and the DWBC, indicating
strong variability there. Periods of max power spectra density (PSD) indicate the dominant feature of the variability
at the large EKE areas is an 80- to 120-day oscillation, based on spectrum analyze of zonal and meridional velocity
time series from the 28th to 29th layers at each gird point for the control run (Fig. 8). This oscillation also presents
in the time series recorded by the six current-meter moorings M1-M6 deployed off the eastern slope of the Zhongsha
Islands (Zhou et al., 2017). The relative leading time between the two closed cells in zonal and meridional directions
can be obtained by calculating the lag correlation of zonal and meridional velocity time series, respectively.
Dividing the corresponding distance, we obtain the mean phase speed and direction of the deep oscillation (Fig. 7b).
The waves show a northwestward propagation in both the deep northeastern circulation and the DWBC, with a
velocity amplitude of ~1.0 to 1.5 cm s$^{-1}$ (Fig. 7b), comparable with the mean speed of ~2.9 cm s$^{-1}$ along the section
M1-M6 (Zhou et al., 2017). Based on the principle axis variance ellipse of band-passed velocity and propagation
direction, Zhou et al. (2017) suggested that the 80- to 120-day oscillation cannot be attributed to topographic Rossby
waves, a mechanism for abyssal intraseasonal variability, especially at the deep western boundary (e.g., Thompson,
1977; Johns and Watts, 1986; Pickart and Watts, 1990; Hamilton, 2009). Other possibilities include the barotropic
and baroclinic Rossby waves. In another sensitivity experiment we doubled the SCS basin and the 80- to 120-
dayoscillation peak disappeared, indicating this oscillation maybe related to the basin mode of the SCS (e.g.,
Platzman, 1972; Xu et al., 2007). This variability is a good topic for future studies.

### 3.4 Model Sensitivity to Distribution of Mixing

Exp-5, Exp-3, Exp-3A, and Exp-3C all show a basin-scale cyclonic gyre with a western intensification in the deep
SCS (Fig. 9). However, the volume transport of the deepwater overflow in the Luzon Strait, the DWBC, and the
detail structure of the deep circulation are quite different in these experiments. The simulated deep circulation is
much weaker in Exp-5 and Exp-3A (e.g., 0.9 and 1.0 Sv is smaller than the control run (1.2 Sv) of the overflow; 1.0
and 0.7 Sv are nearly two times smaller than the control run (2 Sv) at 16.5 ̊N of the DWBC). On the other hand, it is



closer to the control run in the Exp-3 and Exp-3C (1.4 and 1.2 Sv of the overflow; 2.2 and 1.9 Sv of the DWBC).
Magnitude of upwelling is similar case: The upwelling transports southward from 16.5 °N in Exp-5 and Exp-3A (0.6
and 0.6 Sv), two times smaller than the control run (1.2 Sv), while the control run, Exp-3 and Exp-3C are in
reasonable agreement (1.2, 1.3 and 1.1 Sv). This indicates that compared with the north shelf of the SCS with the
Luzon Strait, deep circulation in the SCS is more sensitive to the large vertical mixing parameters of the Zhongsha
Island Chain area. This might be explained by the fact that the latter contains more areas of density difference, as the
deep circulation is essentially density driven. With an increase in the range of strong mixing, the intensity of the
deep circulation in the SCS is enhanced, suggesting that enhanced mixing in the SCS and the Luzon Strait plays an
important role in maintaining the intensity of the SCS deep circulation. At the same time, the spatial structure of the
deep circulation in the SCS also changes. For example, the southwest sub basin circulation is expanded in Exp-5,
while the recirculation near the DWBC extends to the Zhongsha Island Chain area in the control run but not in the
other four experiments. By adjusting the thermohaline structure, enhanced mixing not only impacts the local deep
circulation, but can also influence the deep circulation in other areas without enhanced mixing.
**4. Summary and Discussion**
Due to enhanced mixing in the deep SCS, the deep water in the SCS is expected to move upward much faster than
deep water in the open ocean (on the order of 0.1 cm day$^{-1}$; e.g., Kunze et al., 2006). Qu et al. (2006a) gave an
estimate of area-averaged vertical upwelling velocity of the deepwater in the SCS at $\omega=Q/A$=0.24 m d$^{-1}$, and applied
a hydraulic theory to estimate the Luzon Strait transport $Q$=2.5 Sv and the area of the SCS at 2000 m to estimate as
$A = 9 \times 10^{11}$ m$^2$. Based on long-term mooring observations, the upwelling velocity becomes 0.08 m d$^{-1}$ while $Q$=0.8
Sv (Zhou et al., 2014; Zhao et al., 2016) in this way. Yang et al. (2016) obtained the vertical velocity as 0.32 m d$^{-1}$
from a vertical advective-diffusive balance model based on the diffusivity results inferred from the Gregg-Henyey-
Polzin parameterization and 0.28 m d$^{-1}$ from a dynamically and kinematically consistent ocean state estimate system
(Estimating the Circulation and Climate of the Ocean, ECCO; Forget et al, 2015). For the horizontal distribution of
upwelling in the deep SCS basin, albeit without estimating the magnitude, Shu et al. (2014) indicated there are three
northwest-southeast tilted zones where tracers upwell inferred from the modeled trajectories. These correspond to
the three deep meridional overturning circulation cells. They speculated that one possible mechanism for these
upwelling zones is the interaction between the topographically trapped waves on the slope and the westward



planetary Rossby waves (e.g., Rhines, 1970; Anderson and Gill, 1975).
As described in Fig. 6d, the net transport of the 28th and 29th at these four sections are all southward, with the
values decreasing as 1.25, 1.06, 0.77 and 0.42 Sv, respectively. This indicates the deep flow goes upward from the
deep layer as a result of enhanced mixing in the deep SCS. By dividing the differences between the net transports
with corresponding areas, the upward transports are found to be 0.19, 0.29, 0.35 and 0.42 Sv, which indicate the
values of upwelling at each area are 0.19, 0.32, 0.27 and 0.22 m d$^{-1}$, respectively. We also cumulated the mean
transports along four meridional sections (1.15 Sv at 118.5 °E, 0.88 Sv at 117.0 °E, 0.65 Sv at 115.5 °E and 0.29 Sv at
114.0 °E) and the corresponding upwelling became 0.28, 0.23, 0.36 and 0.29 m d$^{-1}$, respectively. This suggests that
the DWBC is the strongest upwelling area. In order to present the horizontal distribution and magnitude of
upwelling, we cumulated the diapycnal water mass transformation across the upper interface of the 28th layer for the
control run in each $1°\times1°$ box (Fig. 10). The upward transformation is due to interior diapycnal mixing and
elevations around the DWBC and seamounts areas with values of 1 m d$^{-1}$ or larger, while downwelling exists in the
relatively flat inner basin with values of 0.5 m d$^{-1}$. The magnitude of total diapycnal transformation of the SCS (1.5
Sv) is close to that of the deepwater overflow in the Luzon Strait, which means the model drifting is small. Recent
studies indicated that the deep upwelling near the deep west boundary and seamounts may also be driven by near-
boundary mixing (e.g., Ferrari et al., 2016; Mcdougall and Ferrari, 2017).
In the present study, the deep circulation in the SCS is investigated by eddy-resolving model simulations, and
found to be in reasonable agreement with mooring arrays. Analysis of these results provides a detailed structure and
variability of the deep circulation in the SCS. The major features of the SCS deep circulation are a basin-scale
cyclonic gyre and a western intensification. The transport of the DWBC is ~2 Sv at 16.5 °N with a width of ~53 km.
Flowing southwestward, the DWBC becomes weaker and gets a wider range. By dividing the differences between
transports with corresponding areas, the values of upwelling are from 0.19 to 0.36 m d$^{-1}$, with the strongest area
being around the DWBC. The model results reveal the existence of an 80- to 120-day oscillation in the deep
northeastern circulation and the DWBC, which are also the large mean EKE areas. This intraseasonal oscillation has
a northwestward direction, with a velocity amplitude of ~1.0 to 1.5 cm s$^{-1}$ in zonal and meridional velocity. The
distribution of mixing parameters in the deep SCS plays a role in both the spatial structure and volume transport of
the deep circulation. Comparing the northern shelf of the SCS with the Luzon Strait, deep circulation in the SCS is
more sensitive to the large vertical mixing parameters in the Zhongsha Island Chain area. Even though the model is



idealized, the model current fields qualitatively reproduce the results of direct current measurement. The success of
the present model may be associated with several intrinsic features of the deep circulation.
**Data availability**
Model outputs are available upon request to the first author.
**Author contribution**
All the authors conceived and designed the experiments and contributed ideas in the writing process. X.Z.
performed the experiments, analyzed the data and wrote the paper.
**Acknowledgements**
This work was supported by the National Key Research and Development Program of China (Grant no.
2016YFC1402605), the National Natural Science Foundation of China (Grant nos. 41676011, 41806031,
41606014, 91628302), the National Key Research and Development Program of China (Grant no.
2018YFC1407002, 2016YFC1402103), the East Asia Marine Cooperation Platform (China-ASEAN marine
cooperation fund), the National Key Basic Research Program of China (Grant no. 2014CB745003), the Global
Change and Air–Sea Interaction Project (Grant nos. GASI-IPOVAI-01-03, GASI-IPOVAI-01-02), the Foundation
for Innovative Research Groups of the National Natural Science Foundation of China (Grant no. 41521091), the
Key Research and Development Program of Shandong (Grant no. 2016CYJS02A03), and the NSFC-Shandong
Joint Fund for Marine Science Research Centers (Grant no. U1406401).

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





Table 1. Mooring configurations with mean zonal and meridional velocities in different depths.

| Mooring ID | Longitude [°E] | Latitude [°N] | Water depth [m] | Current meter depth [m] | $\overline{U}$ [cm s$^{-1}$] | $\overline{V}$ [cm s$^{-1}$] |
|---|---|---|---|---|---|---|
| M1 | 114°35.761' | 15°14.855' | 3560 | 1940 | -0.47 | -0.07 |
| | | | | 2440 | -1.11 | -0.39 |
| | | | | 2940 | -1.14 | -1.08 |
| | | | | 3440 | -0.58 | -0.51 |
| M2 | 114°42.094' | 15°11.961' | 4282 | 2062 | -0.15 | -0.22 |
| | | | | 2562 | -0.27 | -0.45 |
| | | | | 3062 | -0.48 | -0.76 |
| | | | | 3562 | -0.64 | -1.21 |
| | | | | 4062 | -0.78 | -1.85 |
| M3 | 115°07.607' | 14°56.235' | 4281 | 2061 | 0.02 | -0.21 |
| | | | | 2561 | 0.22 | -0.28 |
| | | | | 3061 | 0.10 | -0.40 |
| | | | | 3561 | -0.30 | -0.44 |
| | | | | 4061 | -0.27 | -0.58 |
| M4 | 115°20.954' | 14°52.977' | 4200 | 1980 | 0.11 | 0.07 |
| | | | | 2480 | 0.32 | 0.62 |
| | | | | 2980 | 0.44 | 0.76 |
| | | | | 3480 | 0.63 | 0.53 |
| | | | | 3980 | 0.19 | 0.39 |
| M5 | 115°51.996' | 14°50.133' | 4266 | 2046 | -0.53 | 0.23 |
| | | | | 2546 | -0.35 | 0.32 |
| | | | | 3046 | -0.30 | 0.22 |
| | | | | 3546 | -0.16 | 0.03 |
| | | | | 4046 | -0.64 | 0.24 |
| M6 | 116°03.241' | 14°53.750' | 4286 | 2066 | -1.33 | 0.55 |
| | | | | 2566 | -0.96 | 0.42 |
| | | | | 3066 | -1.10 | 0.02 |
| | | | | 3566 | -1.39 | -0.36 |
| | | | | 4066 | -1.80 | -0.73 |





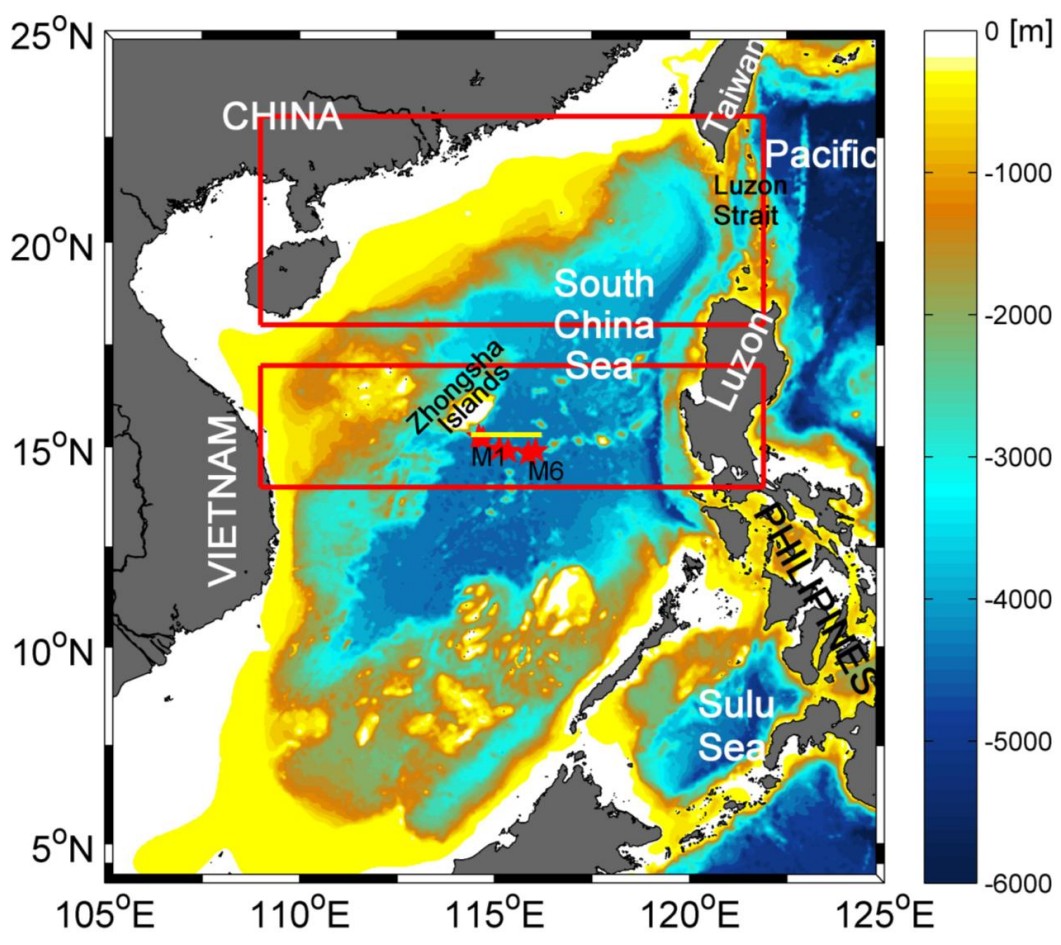


**Figure 1. Bottom topography of the South China Sea. The red stars denote the locations of the year-long mooring array M1-M6. The yellow line indicates the location of model section shown in Fig. 2b. Red boxes indicate the areas with strong mixing in the control run based on Yang et al. (2016).**






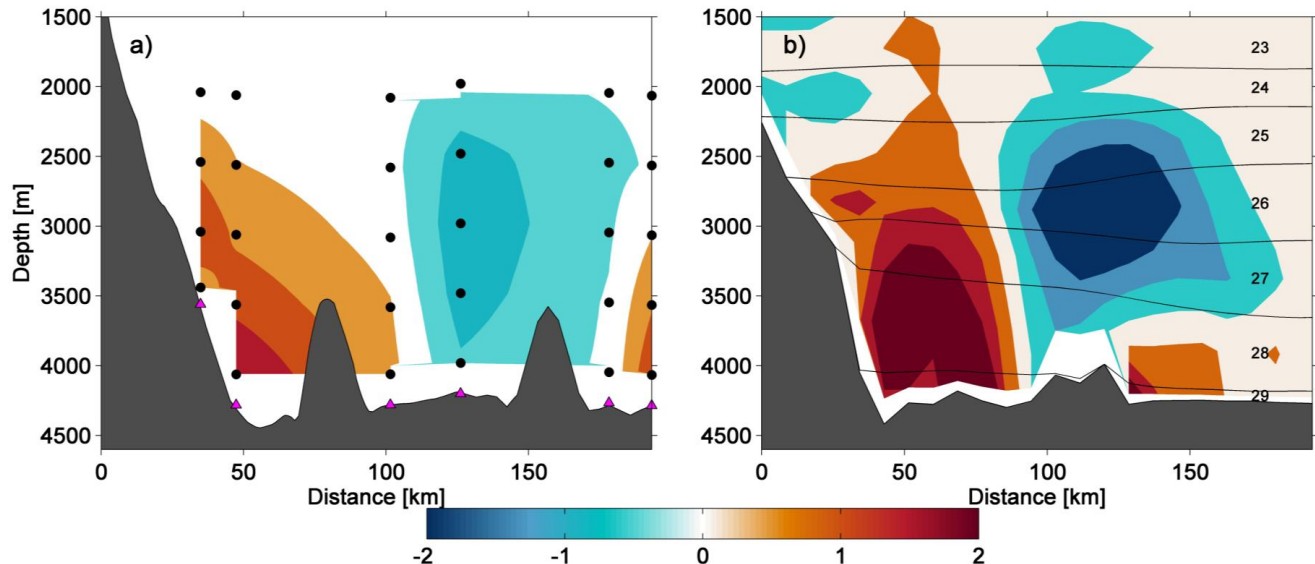


Figure 2. a) Section view of observed mean cross-section velocity (in cm s$^{-1}$) from Zhou et al. (2017; their Fig. 2a). Mooring locations are indicated in magenta triangles.
Locations of current meters are indicated by black dots. b) Time-mean structure of velocity (in cm s$^{-1}$) and thickness numbers at a zonal section of 15.4 °N for the control
run. Note the positive value represents southward velocity.





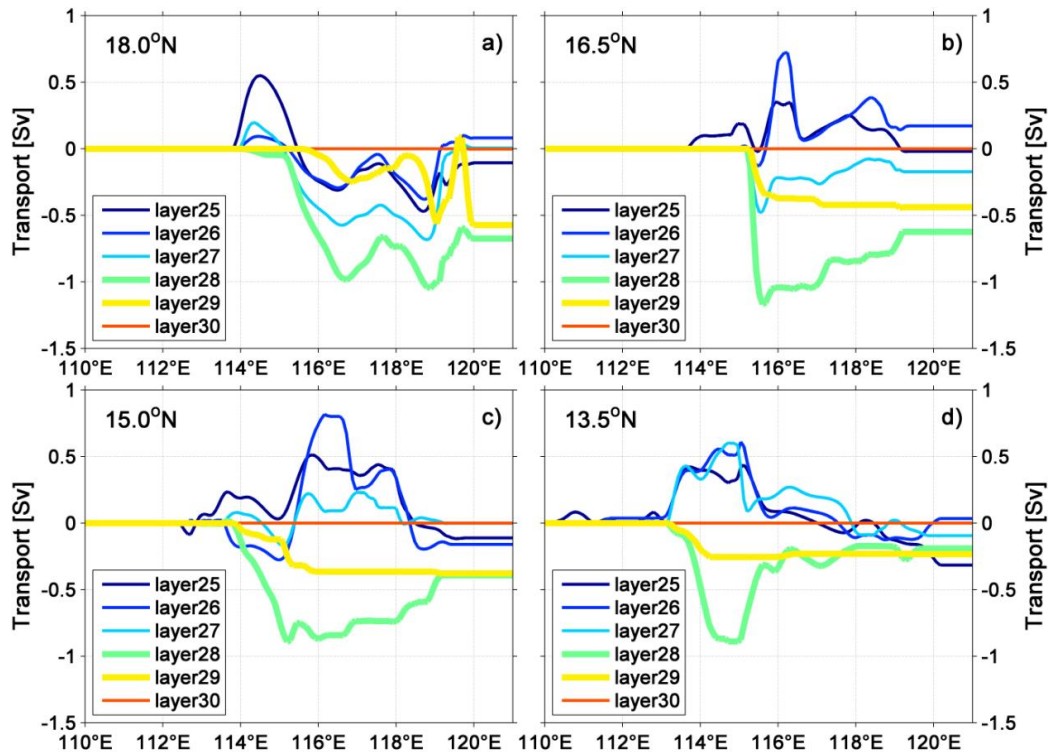


**Figure 3. Eastward cumulated of the meridional volume transports (in Sv) across the model section along 4 zonal sections (13.5 N, 15.0 N, 16.5 N and 18.0 N) of each layer from the 25th to 30th from 110 E to 121 E for the control run. The negative value represents southward volume transport. The depth of the isopycnic interfaces are indicated in Fig. 2b.**




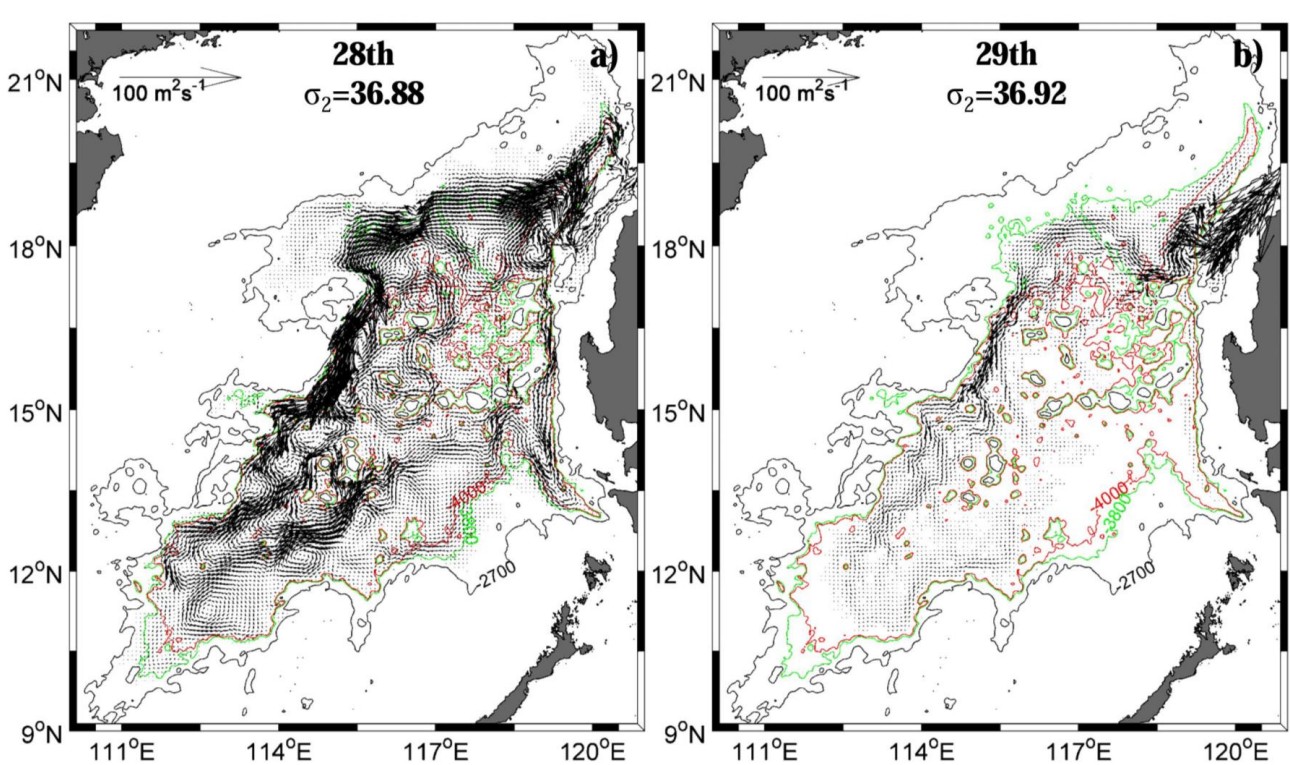

**Figure 4.** Mean volume transport per unit width (in m$^2$ s$^{-1}$) of the 28th (a) and 29th layer (b) for the control run.




**Figure 5. Mean volume transport per unit width (in m² s⁻¹) from the 28th to 29th layer for the control run.**


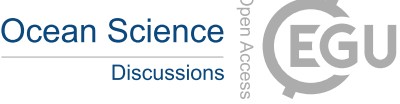



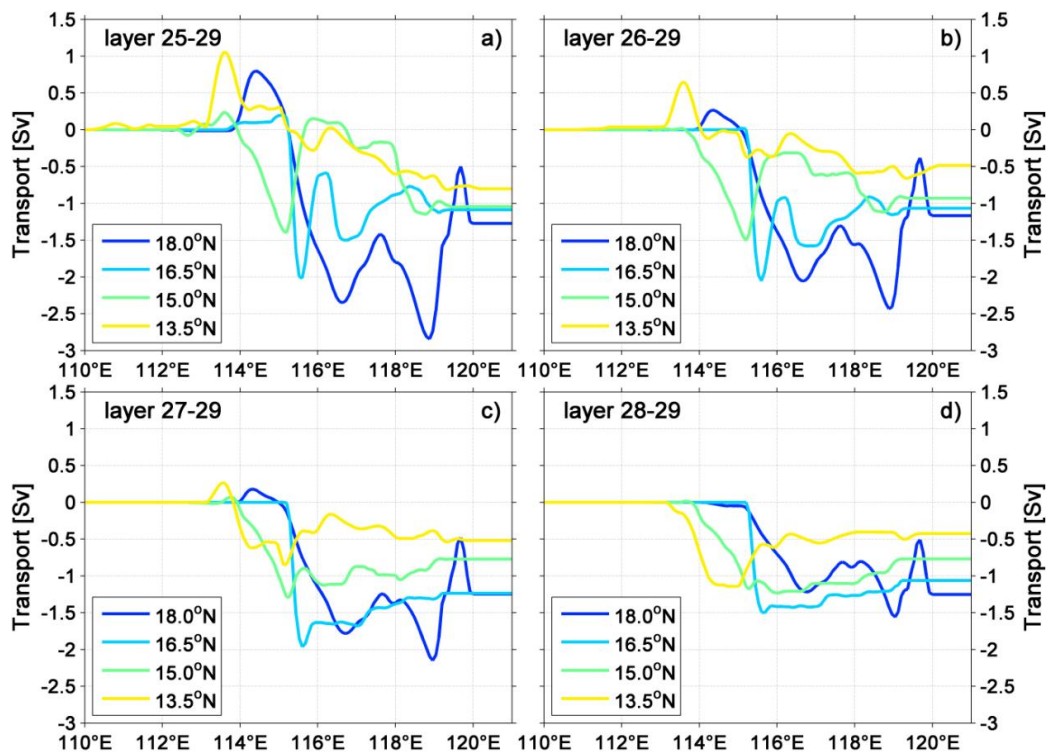


**Figure 6. Eastward cumulated of the meridional volume transports (in Sv) across the model section along 4 zonal sections**
**(13.5 °N, 15.0 °N, 16.5 °N and 18.0 °N) from different layers to 29th from 110 °E to 121 °E for the control run. The negative**
**value represents southward volume transport.**




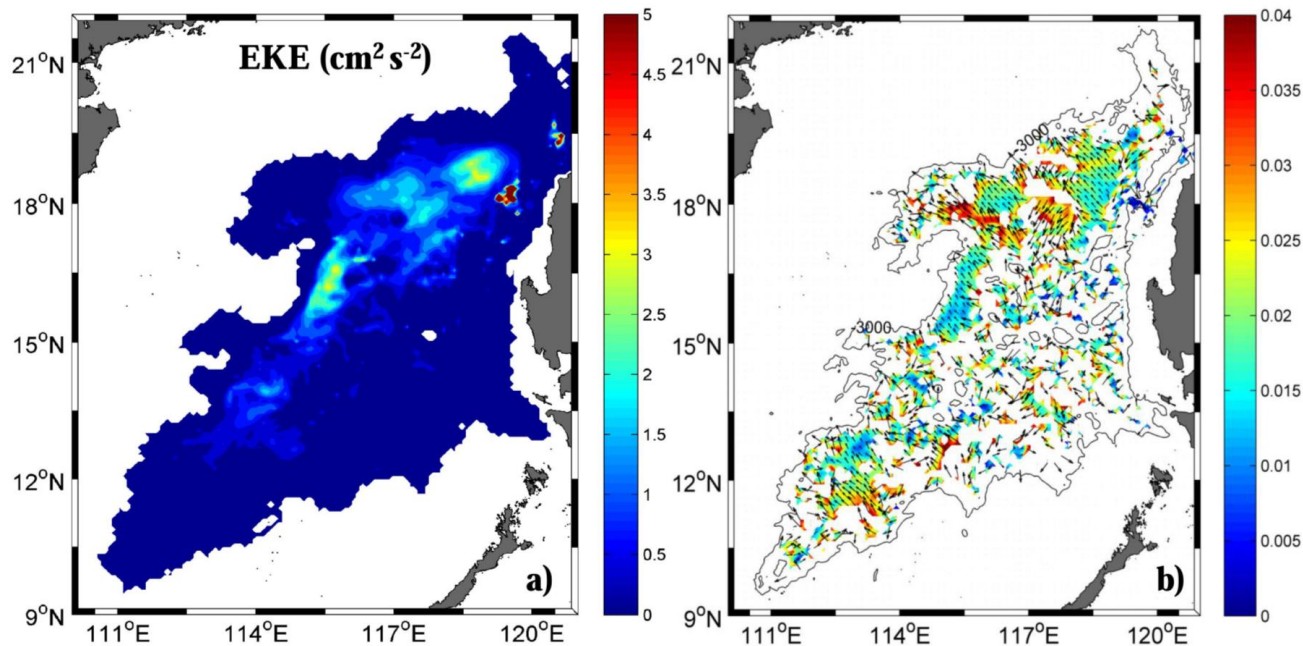

**Figure 7. Distribution of modeled eddy kinetic energy EKE (a, in cm² s⁻²) in the South China Sea, mean phase speed and direction of propagation (b, in m s⁻¹) from the 28th to 29th layer for the control run.**



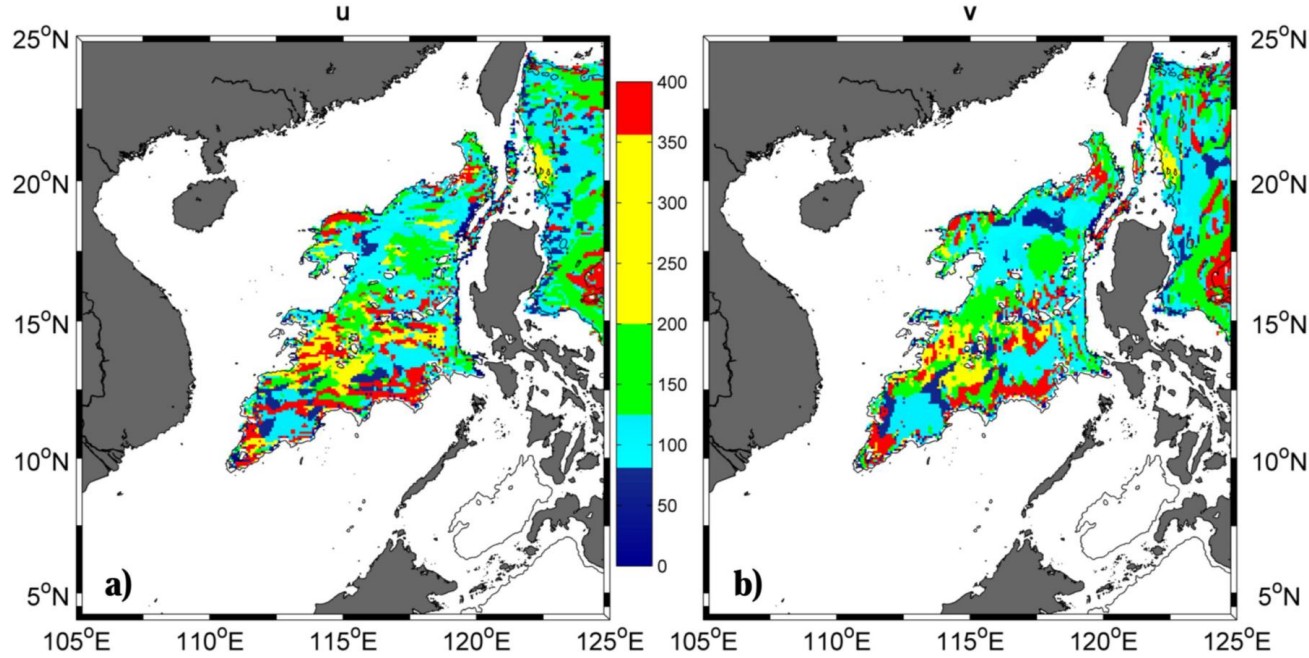


**Figure 8. Periods (in days) of max power spectra density (PSD) of zonal (a) and meridional (b) velocity from the 28th to 29th layer at each gird point for the control run.**



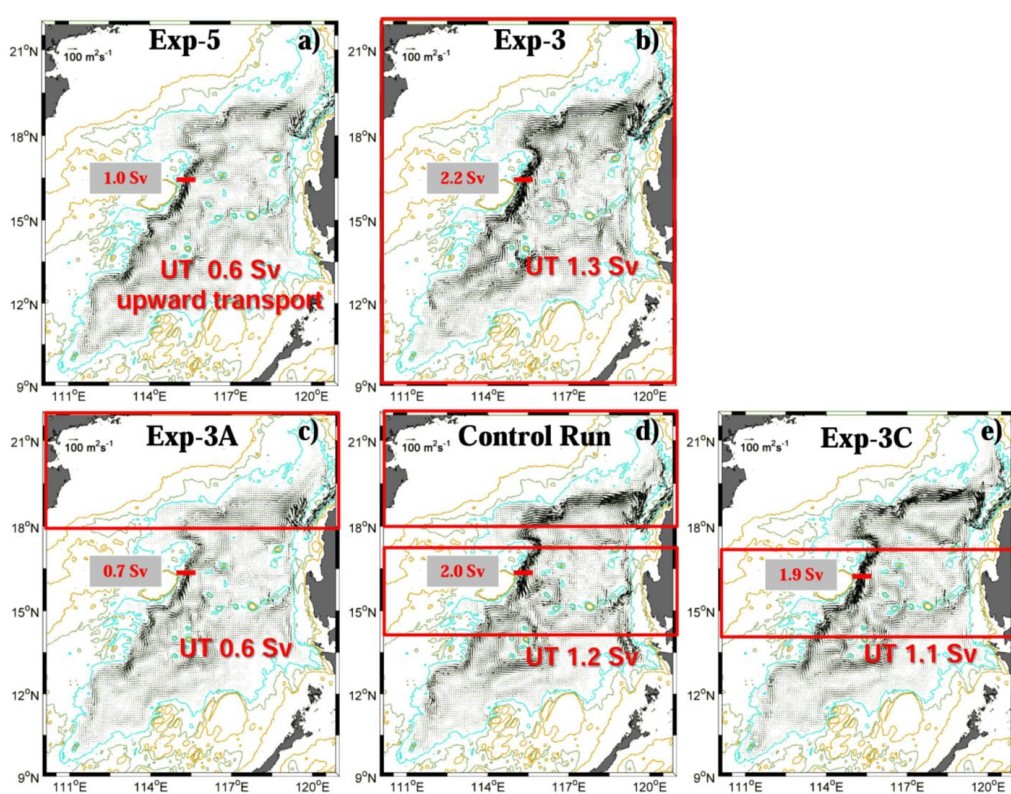


Figure 9. Mean volume transport per unit width (in m² s⁻¹) from the 28th to 29th layer in Exp-5, Exp-3, Exp-3A, control Run, and Exp-3C. The cross sections are indicated by red lines and the corresponding volume transports (in Sv) are indicated in the textboxes with gray background. Red boxes indicate the areas with strong mixing.

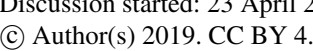
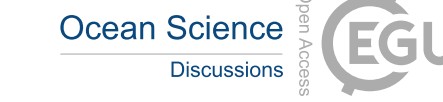


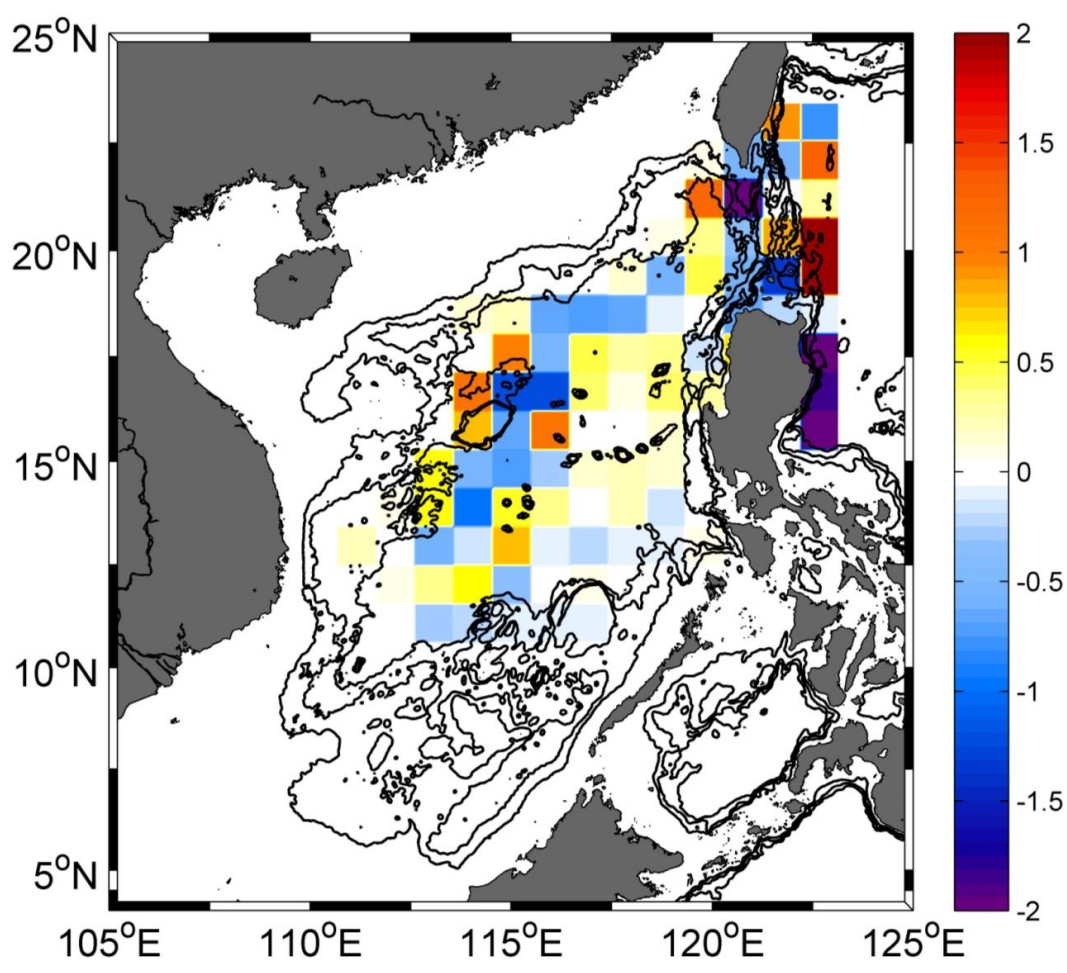


**Figure 10. Horizontal distribution of diapycnal water mass transformation (in m d⁻¹) binned in 1 °×1 ° cells across upper**
**interface of the 28th layer for the control run.**