# Peer review of "Deep Circulation in the South China Sea Simulated in a Regional Model"

_Ocean Science, 2019_

## Referee Comment (RC1) · Anonymous Referee #1 · 14 May 2019

This study uses observation and high resolution simulation to study the deep circulation in the South China Sea. The topic is very important and interesting. This study indeed provides some useful aspects to describe the deep circulation from modeling perspective, especially related to the magnitude of the circulation and potential mechanisms. I think this study would very much interest the research community of South China Sea and I would recommend this study after its revision. I do have some major/medium comments that the authors need to consider very carefully in clarifying some confusing points, only after which this study may be considered to be accepted in Ocean Science.

1. line 106: "Given the lack of observations and inadequate quality control, detailed structures of circulation in the deep SCS have not been mapped out and described adequately"

[Figure]

You have spent many paragraphs and great details introducing previous works on observations and modeling of SCS circulation. But now you suddenly say this circulation "have not been mapped out and described adequately". This statement is too general and may not be fair to previous studies. Please be *specific* about the problems you want to address, and how they are new from previous studies. This is very important.

2. line 107: "Combining the mooring array in Zhou et al. (2017) with results from eddy-resolving model simulations, the present study investigates deep circulation under enhanced mixing in the SCS."

Again, this is too general. Have not any previous studies also used observation and models to "study investigates deep circulation under enhanced mixing in the SCS"? You need to be specific about the new points of your study to distinguish from previous studies.

3. line 129: "Despite the fact that surface forcing is significant in this region as regulating the upper layer circulation, evidence of surface forcing to the deep layer dynamics has not yet been found. Since the current work is designed to be a process study, surface forcing was not applied in the experiments."

"Evidence of surface forcing to deep layer dynamics has not yet been found" does not justify that surface forcing is not important. I think there is a big question mark here, and you cannot just skip the surface forcing simply like this without a detailed discussion, with an excuse of being a process study. For example, have you compared your experiment of "no surface forcing" with an experiment with surface forcing to get this conclusion? For example, of course wind and air-sea buoyancy flux may directly regulate deep ocean circulation down to 3-4 km depths such as at Weddell Sea, see this paper: An idealized model of Weddell Gyre export variability. Journal of Physical Oceanography, 44, 1671-1688. 2014. Moreover, you initialize model from observation. But without surface forcing such as wind and air-sea buoyancy flux, how can you continue to maintain a relatively realistic stratification for your ocean simulation?
**OSD**

For example, air-sea buoyancy flux can cause convection (either shallow or deep) that can significantly modify the ocean stratification even at depth and change the ocean circulation, see this paper: On the abruptness of Bølling–Allerød warming. Journal of Climate, 29, 4965–4975. 2016, and this paper: Ocean Convective Available Potential Energy. Part II: Energetics of Thermobaric Convection and Thermobaric Cabbeling. Journal of Physical Oceanography, 46, 1097–1115. 2016.

My point is this: You may not do further experiment; but I think the authors should discuss the limitations of this configuration of no surface forcing carefully, including at least acknowledge the potential other mechanisms (see papers above) that may have an effect on deep layer dynamics.

4. line 136: "Based on similar configurations with all of the numerical experiments started from rest and integrated for 10 years, Zhao et al. (2014) studied the deep water circulation in the Luzon Strait, which was in good agreement with the observations."

Please be specific on what aspects Zhao et al. (2014) was in good agreement with the observations. If the previous study is already so good, why do you perform the current study? Be specific of what is the new points/motivation of this study comparing with previous study.

5. line 149: "In order to obtain a steady state of the deep circulation in the SCS, we integrated all of the numerical experiments for 20 years"

I know it is expensive to run models. But do you really think 20 years are enough to reach steady state for a new mixing configuration? I thought from textbook it takes 100-1000 years to do so. What is the evidence of a steady state of your simulation? I know it is lots of work, and you may not re-run your model. But please be very careful here and state the limitation sincerely if it exists.

6. line 154: "eddy-resolving model simulations"

You call your $1/12°$ and 32 level model eddy-resolving. Please be careful that this can

be misleading. You should emphasize it is only mesoscale-eddy-resolving but does not well resolve submesoscale eddies (or even smaller-scale turbulence), which can also have great impact on large-scale ocean circulation (see papers below). It may be helpful to discuss/cite these papers: Ocean submesoscales as a key component of the global heat budget. Nature Communications, 9, 775. and Yu et al. 2019. An Annual Cycle of Submesoscale Vertical Flow and Restratification in the Upper Ocean. Journal of Physical Oceanography, doi 10.1175/JPO-D-18-0253

7. Figure 2: this is nice, but why the two panels do not have the same topography?

8. Figure 2: why the simulation has a larger magnitude of velocity than the observation?

9. Figure 2: you put vertical layer number in your simulation result, but why in each layer there is vertical variation of velocity? i.e. how can you have a variation within a grid box?

10. Figure 5: is this figure showing the difference between 28 and 29th layer? not sure what the capture means of "the 28th to 29th layer ". . .please explain.

11. Figure 7: you seem not mention how do you calculate your EKE.

12. Figure 7: you have not explained the EKE patterns carefully in your section 3.3. Why you have large EKE patterns (green, yellow, red) in certain locations? is it due to topography or boundary current? You may want to discuss briefly the role of topography or so on determining EKE, such as shown in this paper: On the Minimum Potential Energy State and the eddy-size-constrained APE Density. Journal of Physical Oceanography, 46, 2663–2674. 2016.

13. Figure 8: why your color scheme is inhomogeneous here? why the yellow range is so large?

14. line 219 "3.4 Model Sensitivity to Distribution of Mixing". You say in line 89 "Yang et al. (2016) recently obtained the three-dimensional distribution of turbulent mixing in the SCS for the first time." Have you tried to use this three-dimensional distribution of

turbulent mixing for your simulation? so you may get more realistic simulation?

15. line 200: "3.3 Temporal Variability of the Deep Circulation". You have not discussed the seasonal variability here. How large is the seasonality? Note the eddy KE and the KE of internal waves can have strong seasonality over the globe including SCS (see paper below). You may may discuss/cite this: Partitioning Ocean Motions Into Balanced Motions and Internal Gravity Waves: A Modeling Study in Anticipation of Future Space Missions, Journal of Geophysical Research, 123, 8084–8105. 2018.

16. last section: you summarize your result here. But you need also to compare your result with previous studies and clearly state what are some new results distinguished from previous studies.

17. last section: You discuss lots about effects of vertical mixing for the circulation, which is still useful although relatively well known. Personally speaking, I think other potential good direction for future studies may include to research on how topography (such as beta effect due to topography), bottom drag, and eddies influence the deep ocean circulation, stratification, and tracer transport. For example, you may take advantage of your high-resolution model to discuss how eddies at different scales (e.g. small scales resolved by high-reso model) may influence ocean circulation, such as by inverse cascade of kinetic energy: good to mention the following study: Klein et al. 2019. Ocean Scale Interactions from Space. Earth and Space Science, AGU, doi 10.1029/2018EA000492 (e.g. see its figure 13)

Minor comments.

line 24: "The Taiwan Strait to the East China Sea in the north", should decapitalize "The"

line 128: what is "1' resolution"?
* * *

---

## Referee Comment (RC2) · Anonymous Referee #2 · 29 May 2019

Review of "Deep Circulation in the South China Sea Simulated in a Regional Model"

Zhao et al. presented a numerical study on the spatial characteristics of the deep western boundary currents (WBCs) in the South China Sea (SCS), using an eddy-resolving configuration of HYCOM. They addressed the role of Pacific overflow water and enhanced diapycnal mixing in studying the dynamics of the WBCs. The study provides some quantitative model diagnosis which could be useful to the relevant research community, however, I am not convinced that this work brings further insight into understanding the dynamics of WBCs in general or specifically in the SCS, and therefore has limited contribution to advancing the science for this topic. The simulations are more like a model exercise on idealized vertical mixing, although the authors have been trying to compare their results to observations and previous model stud-

ies. Many numbers quantified and conclusions drawn seem to merely confirm what has been found before in the literature. In my opinion, this work to some extent also suffers from flawed methodology, inadequate discussion/analysis, missing information, and poor presentation of figures.

I acknowledge the efforts that the authors have put and the potential value of the topic, but I have to recommend rejection of the manuscript with its current status. Some overall and more specific comments are as follows.

1) model configuration (method): I am not convinced that the authors can make any meaningful quantification of the WBCs, nor any one-to-one comparison with observations, without surface forcing and with closed open boundaries. ". . . evidence of surface forcing to the deep layer dynamics has not yet been found" is not a good argument, and does not necessarily mean that surface forcing is not important. It is possible that the role of the surface forcing is not as important as the overflow inflow at Luzon Strait (LS), but it remains unknown unless some sensitivity experiments are carried out to test it. Similarly, ocean circulations in the LS and SCS are highly interactive with circulations in the western Pacific. Although T/S are relaxed at the open boundaries, but without inflow from the Equatorial Pacific and outflow leaving the model domain in the north (e.g. Kuroshio), how would this affect the analysis and quantified properties of the deep WBCs in this work?

If too much work for implementing or turning on the surface forcing and open boundaries, the authors should at least make clear of the limitations in the manuscript, and/or discuss how this could potentially affect their simulation results.

Furthermore, I am also not convinced by the approach the authors did with enhanced mixing. a) The configuration of enhanced mixing is still highly idealized compared to the observations compiled by Yang et al. (2016), which showed an inhomogeneous distribution of diapycnal mixing in the zonal, meridional, and vertical directions. For example, the vertical mixing in the northern SCS and near the bottom is much larger

(up to 10ˆ-2). I am not saying the model setup is wrong, but I think the authors should add more discussions on this point, and make clear of the caveat and idealized nature of their model setup. b) the two hot spots with enhanced mixing, separated by a narrow band of low mixing in the model seems bizarre to me; would it make any difference if this narrow band is also filled with high mixing? There actually seems a lack of observations within this band in the observations of Yang et al.

2) Model spin-up and validation I do not see sufficient validation and assessment of the simulation. A comparison of velocity cross-section with observations (e.g. Fig. 2 only) provides very limited support on the robustness of the simulation results. For instance, how does the model perform in simulating the T/S properties in general for the SCS (at least for the mid- and deep ocean, if not for the whole depth)? Importantly, as the authors deem the dense overflow at the LS to be a crucial factor in determining the dynamics of the WBCs, how is the model behaving in simulating the overflow at the strait? e.g. are the T/S/rho properties of the overflow captured? Are they descending to the bottom basin without being (numerically) diffused? The authors mentioned some numbers on the volume transport of the overflow, but how is the overflow defined?

Besides, regarding the model-data comparison (Fig. 2), "As expected, the control run shows reasonable agreement with the cross-section observations" (L171) seems like an overstatement to me. The WBCs and the return flow are clearly stronger with a broader core compared to the observations. This inconsistency and the possible reasons are not adequately addressed.

As for the model spin-up, the authors need to justify that 20 years are long enough for the model to reach equilibrium or quasi-equilibrium. Evidence should be shown; examples are timeseries of volume transport of the Pacific overflow and the deep WBCs, timeseries of deep basin T/S, and so on.

——————————————————————

Some more specific (not necessarily minor) comments are listed below.
> L9: the first sentence does not need to and should not appear in the abstract;

> L35-37: just to clarify, the 'three-dimensional circulation' is one component of the SCS throughflow, and it is the latter that serves as a heat/freshwater conveyor, right?

> L38-52: while the topic of this work is on the SCS deep WBCs, the authors use lots of space introducing the overflow at the LS (I do understand that the WBCs are strongly linked with the overflow). The authors could consider reducing this part.

On the other hand, having given so much background information on the overflow, the authors are expected to describe a bit more on the model representation of overflows, other than merely giving a number of volume transport. See also my earlier comment 2).

> L41-44: the geographical locations are challenging to the readers. The authors could consider avoiding this, or provide a map showing their locations.

> L46: rms – full name should be given here.

> L68-69: '. . . and higher resolution' – how much higher? e.g. slightly higher than 0.4/0.5 degree, or as high as the resolution in this study?

> L71: what is a/the 'north deep circulation'?

> L76: "Since the DWBC is due to the LS overflow and the beta effect" – this assertion appears to come out of the blue; is this very well known already? Any reference?

> L77: To me the resolution is a secondary consideration. Whether the model produces sufficient overflow water that spills over the sill and whether the water can descend down to the bottom with proper entrainment en route are more important. This could well be my personal biased view though.

> L81: I don't see the logical connection between "Due to the lack of field observations" and the rest of the sentence.

> L85: It should be made more clear that the measurements of Tian et al. and Alford et al. covers the LS and only a small part of the SCS.

> L100: what do you mean by saying "... would have a negative effect on driving the cyclonic SCS deep circulation"?

> L137: please be more specific over here in terms of what is in good agreement with observations. Also, if the model by Zhao et al. (2014) is already performing well in the LS, how about the behavior of the control simulation with enhance mixing at LS?

> L146: larger not lager

> L153: "simulated"

> L154: "mechanism" – I don't see much analysis or discussion on the mechanism of the deep WBCs throughout the main text.

> L158-160: I don't really understand what the authors mean by "re-coordination" and "projection" here.

> L164-166: to me this seems too speculative.

> L170-171: what do the authors mean by "is the same status"? do they mean that the model reproduces the observed volume transport of the overflow, but overestimates the strength of the deep WBCs?

> Figure. 2: a comparison of cross-sectional T/S/rho would also be beneficial.

> L174: across, not along?

> Figure. 3: It seems like this figure is not sufficiently discussed/cited in the main text, which questions the inclusion of all the four panels in the figure.

> Figure. 4: the figure needs to be improved. 1) The vectors are crowdedly clustered in the WBC region, whereas in the other region the arrow heads (aka directions of flow) are very difficult to see. 2) Are the three isobaths randomly selected? 3) units

are missing for the value of sigma_2; "28th layer" not just "28th". 4) the authors could consider making the plot with data from, say every 5th grid point, to avoid the busy vectors, or could consider plotting the stream function.

> L182: "Therefore, . . ." I don't see the logic of 'therefore' here.

> L188-192: the description on the WBC path is challenging for the readers to follow; please consider adding some schematic arrows in the figure.

> Figure. 5: I don't see the point of having both Figures 4 and 5 in the paper.

> Figure. 6: this figure just seems to be a variant of Figure. 3. I don't see much added value or information from it.

> L196-197: not necessarily; strong entrainment is expected to occur during the descent of overflow water.

> L202: how is EKE defined? It might be obvious for some, but not to all readers.

> L216-218: This seems a weak indication to me without knowing more key information from this sensitivity experiment. For instance, is the strength and distribution of the overflow and/or WBCs similar to that in the control experiment?

> Figure. 9: this is a very poor figure. Perhaps a short table would summarize what the authors want to say over here; but if they do would like to show the figure, please make sure that it is displayed clearly and is readable.

> L223-224: why is the intensity of the deep WBCs in case Exp-3A reduced relative to Exp-5? Could it be because more dense waters from the lower layers (e.g. 28 and 29) are mixed upwards due to the enhanced mixing in the northern SCS? If yes, perhaps more layers should be included for a fair comparison? The same applies to the other experiments. Furthermore, this might be too much to ask, how does the deep ocean T/S change with enhanced/reduced vertical mixing?

> L230-231: I don't understand this sentence here.

> Figure. 10: If I understand correctly, there are negative values (i.e. downwelling) along the deep WBC path (e.g. east and south of Zhongsha Islands). This seems to be contrary to what the authors claim in the text?

―――――――――――――――――――――――

---

## Author Comment (AC1) · 20 Aug 2019

Thank you for the constructive comments and suggestions regarding our manuscript "*Deep Circulation in the South China Sea Simulated in a Regional Model*" [paper No.: os-2019-29]. The revised manuscript is attached. All the comments have been considered. Below are the detailed responses (in black) to the reviewer's comments (in blue).

**Anonymous Referee #1**

This study uses observation and high resolution simulation to study the deep circulation in the South China Sea. The topic is very important and interesting. This study indeed provides some useful aspects to describe the deep circulation from modeling perspective, especially related to the magnitude of the circulation and potential mechanisms. I think this study would very much interest the research community of South China Sea and I would recommend this study after its revision. I do have some major/medium comments that the authors need to consider very carefully in clarifying some confusing points, only after which this study may be considered to be accepted in Ocean Science.

[#1] Referee Comment: line 106: "Given the lack of observations and inadequate quality control, detailed structures of circulation in the deep SCS have not been mapped out and described adequately".

You have spent many paragraphs and great details introducing previous works on observations and modeling of SCS circulation. But now you suddenly say this circulation "have not been mapped out and described adequately". This statement is too general and may not be fair to previous studies. Please be *specific* about the problems you want to address, and how they are new from previous studies. This is very important.

Author's response: Thanks for this suggestion. We agreed this statement is too general. What we want to address is the sensitivity of the SCS deep circulation to the observed distribution of mixing with two mixing "hotspots", as previous numerical studies simulated the deep circulation with homogeneous or simulated vertical mixing parameters in the deep SCS. After reconsideration, the upper and lower paragraphs of this sentence have clarified this point. So we remove this sentence from the manuscript.

Changes in manuscript: Description corrected. (lines 112-113)

[#2] Referee Comment: line 107: "Combining the mooring array in Zhou et al. (2017) with results from eddy-resolving model simulations, the present study investigates deep circulation under enhanced mixing in the SCS."

Again, this is too general. Have not any previous studies also used observation and models to "study investigates deep circulation under enhanced mixing in the SCS"? You need to be specific about the new points of your study to distinguish from previous studies.

Author's response: Zhao et al. (2014) has studied the impact of enhanced mixing on the deep overflow through the Luzon Strait. But inside the deep SCS, to the best of our knowledge, no previous studies have investigated the regulation of enhanced mixing on the deep circulation yet. So the sentence is modified to be specific about this.

Changes in manuscript: Description corrected. (lines 113-115)

[#3] Referee Comment: line 129: "Despite the fact that surface forcing is significant in this region as regulating the upper layer circulation, evidence of surface forcing to the deep layer dynamics has not yet been found. Since the current work is designed to be a process study, surface forcing was not applied in the experiments."

"Evidence of surface forcing to deep layer dynamics has not yet been found" does not justify that surface forcing is not important. I think there is a big question mark here, and you cannot just skip the surface forcing simply like this without a detailed discussion, with an excuse of being a process study. For example, have you compared your experiment of "no surface forcing" with an experiment with surface forcing to get this conclusion? For example, of course wind and air-sea buoyancy flux may directly regulate deep ocean circulation down to 3-4 km depths such as at Weddell Sea, see this paper: An idealized model of Weddell Gyre export variability. Journal of Physical Oceanography, 44, 1671-1688. 2014. Moreover, you initialize model from observation. But without surface forcing such as wind and air-sea buoyancy flux, how can you continue to maintain a relatively realistic stratification for your ocean simulation? For example, air-sea buoyancy flux can cause convection (either shallow or deep) that can significantly modify the ocean stratification even at depth and change the ocean circulation, see this paper: On the abruptness of Bølling–Allerød warming. Journal of Climate, 29, 4965–4975. 2016, and this paper: Ocean Convective Available Potential Energy. Part II: Energetics of Thermobaric Convection and Thermobaric Cabbeling. Journal of Physical Oceanography, 46, 1097–1115. 2016.

My point is this: You may not do further experiment; but I think the authors should discuss the limitations of this configuration of no surface forcing carefully, including at least acknowledge the potential other mechanisms (see papers above) that may have an effect on deep layer dynamics.

Author's response: Thanks for the suggestions and the references. We discussed about this and acknowledge the limitations of this configuration in the manuscript.

Changes in manuscript: We rephrased the sentence and discussed the limitations of no surface forcing in the model configuration of the manuscript. (lines 136-138, 299-306)

[#4] Referee Comment: line 136: "Based on similar configurations with all of the numerical experiments started from rest and integrated for 10 years, Zhao et al. (2014) studied the deep water circulation in the Luzon Strait, which was in good agreement with the observations."

Please be specific on what aspects Zhao et al. (2014) was in good agreement with the observations. If the previous study is already so good, why do you perform the current study? Be specific of what is the new points/motivation of this study comparing with previous study.

Author's response: The numerical experiments in Zhao et al. (2014) was in good agreement with their observations of the deep water overflow in the Luzon Strait based on repeated conductivity-temperature-depth (CTD) and lowered acoustic Doppler current profiler (LADCP) surveys. No comparison between simulations and observations inside the SCS was made in their study. Besides, Zhao et al. (2014) simulated the deep circulation with homogeneous vertical mixing parameters in the deep SCS, and one wonders about the sensitivity of the SCS deep circulation to the observed distribution of mixing. Thus, we modified the K-profile parameterization (KPP) mixing scheme in accordance with the two observed mixing "hotspots" found in Yang et al. (2016).

Changes in manuscript: We rephrased the sentence as suggested. (lines 145-146)

[#5] Referee Comment: line 149: "In order to obtain a steady state of the deep circulation in the SCS, we integrated all of the numerical experiments for 20 years".

I know it is expensive to run models. But do you really think 20 years are enough to reach steady state for a new mixing configuration? I thought from textbook it takes 100-1000 years to do so. What is the evidence of a steady state of your simulation? I know it is lots of work, and you may not re-run your model. But please be very careful here and state the limitation sincerely if it exists.

Author's response: Thanks for this suggestion. Fig. RC1 shows the section view of year-mean thickness structure at zonal and meridional section for the control run. The thickness structure was basically stable in the last five years indicated the control run obtained a steady state of the deep circulation in the SCS.

[Figure]

**Figure RC1. Section view of year-mean thickness structure at a zonal section of 16.5 °N (a) and a meridional section of 116 °E (b) for the control run. Thickness numbers and density referenced to 2000 m ($\sigma_2$, kg m$^{-3}$) are indicated.**

Changes in manuscript: We added figure and description to clarify this. (lines 163-164)

[#6] Referee Comment: line 154: "eddy-resolving model simulations"

You call your 1/12_ and 32 level model eddy-resolving. Please be careful that this can be misleading. You should emphasize it is only mesoscale-eddy-resolving but does not well resolve submesoscale eddies (or even smaller-scale turbulence), which can also have great impact on large-scale ocean circulation (see papers below). It may be helpful to discuss/cite these papers: Ocean submesoscales as a key component of the global heat budget. Nature Communications, 9, 775. and Yu et al. 2019. An Annual Cycle of Submesoscale Vertical Flow and Restratification in the Upper Ocean. Journal of Physical Oceanography, doi 10.1175/JPO-D-18-0253

Author's response: Thanks and our model is indeed mesoscale-eddy-resolving. Recent studies show significant impact of submesoscale processes on the overturning circulation and deep circulation. Similar to the internal tides, these processes are basically claimed to interact with topography or other processes and notably modify the stratification through mixing. In this study, we focus on the impact of mixing on the deep circulation. A short discussion on the impact of submesoscale processes on the mixing has been added to the manuscript. It is a big topic to profoundly investigate the dynamics and requires further studies.

Changes in manuscript: We corrected the description to emphasize this. (lines 10, 88-90, 114, 168, 285)

[#7] Referee Comment: Figure 2: this is nice, but why the two panels do not have the same topography?

Author's response: The section topography in the left panel (as Fig. 2a in Zhou et al., 2017) is determined by the connection between the six mooring locations, and the bathymetry data are downloaded from http://topex.ucsd.edu/marine_topo/. With higher resolution that far more than six locations, we decide to present a detailed velocity zonal section view of 15.4°N and used the topography applied to model (from version 13.1 of Smith and Sandwell, 1997) as shown in the right panel.

[#8] Referee Comment: Figure 2: why the simulation has a larger magnitude of velocity than the observation?

Author's response: The simulated DWBC (4 cm s$^{-1}$) and recirculation are stronger than the observations (2 cm s$^{-1}$) is probably due to that the source, deepwater overflow in the Luzon Strait, is the same status (1.2 to 0.8 Sv; Zhou et al., 2014; Zhao et al., 2016).

Changes in manuscript: We noted this in Section 3.1. (lines 183-185)

[#9] Referee Comment: Figure 2: you put vertical layer number in your simulation result, but why in each layer there is vertical variation of velocity? i.e. how can you have a variation within a grid box?

Author's response: Based on the Fig. 2a in Zhou et al. (2017), "The velocity is interpolated and mapped on a finer mesh with horizontal and vertical grid size of 0.1 km and 20 m." To be consistent with them, we used the same method to plot the right panel.

[#10] Referee Comment: Figure 5: is this figure showing the difference between 28 and 29th layer? not sure what the capture means of "the 28th to 29th layer"… please explain.

Author's response: This figure shows the total mean transport per unit width of the 28th and 29th layer.

Changes in manuscript: Description corrected. (lines 196-197)

[#11] Referee Comment: Figure 7: you seem not mention how do you calculate your EKE.

Author's response: Thanks and EKE in this figure is defined as $0.5 \times \left[ (u - \bar{u})^2 + (v - \bar{v})^2 \right]$, where $u$ and $v$ are zonal and meridional velocities, respectively.

Changes in manuscript: We now note this in Section 3.3. (lines 216-217)

[#12] Referee Comment: Figure 7: you have not explained the EKE patterns carefully in your section 3.3. Why you have large EKE patterns (green, yellow, red) in certain locations? is it due to topography or boundary current? You may want to discuss briefly the role of topography or so on determining EKE, such as shown in this paper: On the Minimum Potential Energy State and the eddy-size-constrained APE Density. Journal of Physical Oceanography, 46, 2663–2674. 2016.

Author's response: Thanks for this suggestion. Large EKE areas appear in the deep northeastern circulation and the DWBC can be due to the intricate influences from topography, standing meanders, nonlocal energy propagation, turbulent energy cascade, and so on (e.g., Su and Ingersoll, 2016).

Changes in manuscript: We discussed the potential dynamics for the large EKE areas in the deep northeastern circulation and the DWBC. (lines 218-219)

[#13] Referee Comment: Figure 8: why your color scheme is inhomogeneous here? why the yellow range is so large?

Author's response: Fig. RC2 shows the homogeneous color scheme of new Fig. 9 with a similar pattern. In order to emphasize the different periods in the deep SCS, especially the dominant 80- to 120-day oscillation at the large EKE areas, we used the inhomogeneous color scheme as 0-79, 80-120, 121-200 and 201-365 to highlight these representational periods.

[Figure]

**Figure RC2. Periods (in days) of max power spectra density (PSD) of zonal (left panel) and meridional (right panel) velocity from the 28th to 29th layer at each gird point for the control run.**

[#14] Referee Comment: line 219 "3.4 Model Sensitivity to Distribution of Mixing". You say in line 89 "Yang et al. (2016) recently obtained the three-dimensional distribution of turbulent mixing in the SCS for the first time." Have you tried to use this three-dimensional distribution of turbulent mixing for your simulation? so you may get more realistic simulation?

Author's response: Although the hydrographic measurements in Yang et al. (2016) covered the SCS with a total of 335 stations (477 casts), this three-dimensional distribution of turbulent mixing is still discrete for us to simulation. Thus, we use a continuous turbulent mixing field based on the two mixing

"hotspots" found in Yang et al. (2016) to modify the K-profile parameterization mixing scheme.

[#15] Referee Comment: line 200: "3.3 Temporal Variability of the Deep Circulation". You have not discussed the seasonal variability here. How large is the seasonality? Note the eddy KE and the KE of internal waves can have strong seasonality over the globe including SCS (see paper below). You may discuss/cite this: Partitioning Ocean Motions Into Balanced Motions and Internal Gravity Waves: A Modeling Study in Anticipation of Future Space Missions, Journal of Geophysical Research, 123, 8084–8105. 2018.

Author's response: We agreed the eddy KE and the KE of internal waves can have strong seasonality over the globe including SCS, especially in the upper ocean.

Changes in manuscript: As the simulated seasonal variability of the deep circulation in the SCS is much smaller than the intraseasonal variability at the large EKE areas (see new Fig. 8 and Fig. 9), we rephrased the title of Section 3.3 to emphasize the intraseasonal variability of the deep circulation.

[#16] Referee Comment: last section: you summarize your result here. But you need also to compare your result with previous studies and clearly state what are some new results distinguished from previous studies.

Author's response: On one hand, the location, transport and width of the DWBC are more consistent with the cross-section mooring observations than the previous climate state and model results. On the other hand, the present study for the first time investigates deep circulation under two mixing "hotspots" in the SCS, and open new routes to understand the dynamic that mixing regulating deep circulation.

Changes in manuscript: We added description to clarify this. (lines 297-298)

[#17] Referee Comment: last section: You discuss lots about effects of vertical mixing for the circulation, which is still useful although relatively well known. Personally speaking, I think other potential good direction for future studies may include to research on how topography (such as beta effect due to topography), bottom drag, and eddies influence the deep ocean circulation, stratification, and tracer transport. For example, you may take advantage of your high-resolution model to discuss how eddies at different scales (e.g. small scales resolved by high-reso model) may influence ocean circulation, such as by inverse cascade of kinetic energy: good to mention the following study: Klein et al. 2019. Ocean Scale Interactions from Space. Earth and Space Science, AGU, doi 10.1029/2018EA000492 (e.g. see its figure 13)

Author's response: Topography, bottom drag, eddies, internal tides, submesoscale processes, as well as the surface forcing mention by the reviewer are also all significant factors which have potential impact on the deep circulation directly or indirectly. The reviewer pointed out some constructive suggestions, and we will try to investigate these factors profoundly in the further studies.

Minor comments.

[#18] Referee Comment: line 24: "The Taiwan Strait to the East China Sea in the north", should decapitalize "The"

Author's response: Thank you and corrected. (line 25)

Author's response: The sentence is rewritten as "The bottom topography is from version 13.1 of Smith and Sandwell (1997) with $1/60\,°$ resolution". (line 135)

**Deep Circulation in the South China Sea Simulated in a Regional Model**

Xiaolong Zhao[1,2], Chun Zhou[2], Xiaobiao Xu[3], Ruijie Ye[2], Jiwei Tian[2] and Wei Zhao[2]

[1]North China Sea Marine Forecasting Center, State Oceanic Administration, Qingdao, 266061, P. R. China.
[2]Key Laboratory of Physical Oceanography/CIMST, Ocean University of China and Qingdao National Laboratory for Marine Science and Technology, Qingdao 266100, P. R. China.
[3]Center for Ocean-Atmospheric Prediction Studies (COAPS), Florida State University, Tallahassee, FL, USA.

*Correspondence to:* Chun Zhou (chunzhou@ouc.edu.cn)

**Abstract.**  In this study, deep circulation in the South China Sea (SCS) is investigated using results from mesoscale-
[revised manuscript text omitted]
 one of the reason. Second, different models may have different performances on the entrainment and mixing of ambient water after the deepwater overflow spills into the deep SCS. Third, in most simulations there is a strong cyclonic or anticyclonic circulation cell at the southwest part of the deep circulation under week mixing: a separate cyclonic circulation in Chao et al. (1996) and Shu et al. (2014), while there is an anticyclonic one in Xu and Oey (2014).

Simulation results of the deep circulation in the SCS need to be verified based on observations before being employed to the discussion of the spatio-temporal characteristics of the deep circulation in the SCS.

With progresses on the dynamics of submesoscale processes and internal tides (e.g., Su et al., 2018; Yu et al.,

2019; Polzin et al., 1996), abyssal enhanced mixing generated by these processes and its impact on the stratification and deep circulation has been drawn increasing attention. Enhanced mixing is a well-observed feature in the SCS.

[revised manuscript text omitted]

Strait (109-122 °E, 18-23 °N) and the Zhongsha Island Chain area (109-122 °E, 14-17 °N), respectively. Instead of applying the exact results of mixing distribution of Yang et al. (2016), these configurations are idealized to some extent, in order to reproduce the two mixing "hotspots" dynamically explained by dissipation of internal tides, while not following the specific distribution and magnitude which still need to be verified due to the limitations of numbers of CTD profiles and parameterization method. These configurations may somehow introduce uncertainty to the simulation results which is difficult to evaluate with the current observations.

In order to obtain a steady state of the deep circulation in the SCS, we integrated all of the numerical experiments for 20 years and averaged the last five years as the simulated annual mean results mentioned below (as shown in Fig.

2, the thickness structure was basically stable in the last five years indicated the control run has been stable during the last 10 years).

**3. Key Results**

[revised manuscript text omitted]

2015), is not applied to the numerical experiments. Although configured with a buffer zone near the eastern boundary, the experiments are currently configured with closed lateral boundary condition, which cannot simulate the interactions between the processes of the current model domain and the Pacific/Indonesia seas. These limitations may introduce uncertainty to some extent to the simulation results in this study. The potential impact of surface forcing and boundary conditions on the deep circulation in the SCS is worth to be investigated.

**Data availability**

Model outputs are available upon request to the first author.

**Author contribution**

[revised manuscript text omitted]

6313(59)90075-9, 1960b.

Su, Z., Stewart A., and Thompson A.: An idealized model of Weddell Gyre export variability, J. Phys. Oceanogr.,

44, 1671-1688, doi:10.1175/JPO-D-13-0263.1, 2014.

Su, Z., Ingersoll, A. P., Stewart, A. L., and Thompson, A. F.: Ocean convective available potential energy. Part II:

Energetics of thermobaric convection and thermobaric cabbeling, J. Phys. Oceanogr., 46, 1097-1115, doi:10.1175/JPO-D-14-0156.1, 2016a.

Su, Z., Ingersoll, A. P., and He, F.: On the abruptness of Bølling–Allerød warming, J. Climate, 29, 4965-4975, doi:10.1175/JCLI-D-15-0675.1, 2016b.

Su, Z. and Ingersoll, A. P.: On the minimum potential energy state and the eddy-size-constrained APE density, J.

Phys. Oceanogr., 46, 2663-2674, doi:10.1175/JPO-D-16-0074.1, 2016.

Su, Z., Wang, J., Klein, P., Thompson, A. F., and Menemenlis, D.: Ocean submesoscales as a key component of the global heat budget, Nature Communications, 9(1), 775, doi:10.1038/s41467-018-02983-w, 2018.

Thompson, R. O. R. Y.: Observations of Rossby waves near Site D, Prog. in Oceanogr., 7, 1-28, 1977.

Tian, J., Zhou, L., Zhang, X., Liang, X., Zheng, Q., and Zhao, W.: Estimates of M2 internal tide energy fluxes along the margin of Northwestern Pacific using TOPEX/POSEIDON altimeter data, Geophys. Res. Lett., 30, 1889, doi:10.1029/2003GL018008, 2003.

Tian, J., Yang, Q., Liang, X., Xie, L., Hu, D., Wang, F., and Qu, T.: Observation of Luzon Strait transport, Geophys.

Res. Lett., 33, L19607, doi:10.1029/2006GL026272, 2006.

Tian, J., Yang, Q., and Zhao, W.: Enhanced diapycnal mixing in the South China Sea, J. Phys. Oceanogr., 39, 3191-

3203, doi:10.1175/2009JPO3899.1, 2009.

Tian, J. and Qu, T.: Advances in research on the deep South China Sea circulation, Chin. Sci. Bull., 57, 3115-3120, doi:10.1007/s11434-012-5269-x, 2012.

Wang, J.: Observation of abyssal flows in the Northern South China Sea, Acta Oceanogr. Taiwan, 16, 36-45, 1986.

Wang, G., Xie, S.-P., Qu, T., and Huang, R. X.: Deep South China Sea circulation, Geophys. Res. Lett., 38, L05601, doi:10.1029/2010GL046626, 2011.

Wang, X., Liu, Z., and Peng, S.: Impact of Tidal Mixing on Water Mass Transformation and Circulation in the South

China Sea, J. Phys. Oceanogr., 47, 419-432, doi:10.1175/JPO-D-16-0171.1, 2017.

Wyrtki, K.: Physical oceanography of the southeast Asian Waters, Naga Rep., 2, 195, Scripps Inst. of Oceanogr. San

Diego, Calif, 1961.

Xu, F. and Oey, L. Y.: State analysis using the Local Ensemble Transform Kalman Filter (LETKF) and the three- layer circulation structure of the Luzon Strait and the South China Sea, Ocean. Dynam., 64, 905-923, doi:10.1007/s10236-014-0720-y, 2014.

Xu, Y., Rolph, W. D., Mark, W., and Jae-Hun, P.: Fundamental-mode basin oscillations in the japan/east sea, Geophys. Res. Lett., 34(4), 545-559, 2007.

Yang, J.: Local and remote wind stress forcing of the seasonal variability of the Atlantic Meridional Overturning

Circulation (AMOC) transport at 26.5 °N, J. Geophys. Res. Oceans, 120(4): 2488-2503, doi:10.1002/2014JC010317, 2015.

[revised manuscript text omitted]

---

## Author Comment (AC2) · 20 Aug 2019

Thank you for the constructive comments and suggestions regarding our manuscript "*Deep Circulation in the South China Sea Simulated in a Regional Model*" [paper No.: os-2019-29]. The revised manuscript is attached. All the comments have been considered. Below are the detailed responses (in black) to the reviewer's comments (in blue).

**Anonymous Referee #2**

Review of "Deep Circulation in the South China Sea Simulated in a Regional Model" Zhao et al. presented a numerical study on the spatial characteristics of the deep western boundary currents (WBCs) in the South China Sea (SCS), using an eddyresolving configuration of HYCOM. They addressed the role of Pacific overflow water and enhanced diapycnal mixing in studying the dynamics of the WBCs. The study provides some quantitative model diagnosis which could be useful to the relevant research community, however, I am not convinced that this work brings further insight into understanding the dynamics of WBCs in general or specifically in the SCS, and therefore has limited contribution to advancing the science for this topic. The simulations are more like a model exercise on idealized vertical mixing, although the authors have been trying to compare their results to observations and previous model studies. Many numbers quantified and conclusions drawn seem to merely confirm what has been found before in the literature. In my opinion, this work to some extent also suffers from flawed methodology, inadequate discussion/analysis, missing information, and poor presentation of figures.

I acknowledge the efforts that the authors have put and the potential value of the topic, but I have to recommend rejection of the manuscript with its current status. Some overall and more specific comments are as follows.

[#1] Referee Comment: 1) model configuration (method): I am not convinced that the authors can make any meaningful quantification of the WBCs, nor any one-to-one comparison with observations, without surface forcing and with closed open boundaries. ". . . evidence of surface forcing to the deep layer dynamics has not yet been found" is not a good argument, and does not necessarily mean that surface forcing is not important. It is possible that the role of the surface forcing is not as important as the overflow inflow at Luzon Strait (LS), but it remains unknown unless some sensitivity experiments are carried out to test it. Similarly, ocean circulations in the LS and SCS are highly interactive with circulations in the western Pacific. Although T/S are relaxed at the open boundaries, but without inflow from the Equatorial Pacific and outflow leaving the model domain in the north (e.g. Kuroshio), how would this affect the analysis and quantified properties of the deep WBCs in this work?

If too much work for implementing or turning on the surface forcing and open boundaries, the authors should at least make clear of the limitations in the manuscript, and/or discuss how this could potentially affect their simulation results.

Furthermore, I am also not convinced by the approach the authors did with enhanced mixing. a) The configuration of enhanced mixing is still highly idealized compared to the observations compiled by Yang et al. (2016), which showed an inhomogeneous distribution of diapycnal mixing in the zonal, meridional, and vertical directions. For example, the vertical mixing in the northern SCS and near the bottom is much larger (up to 10^-2). I am not saying the model setup is wrong, but I think the authors should add more discussions on this point, and make clear of the caveat and idealized nature of their model setup. b) the two hot spots with enhanced mixing, separated by a narrow band of low mixing in the model seems bizarre to me; would it make any difference if this narrow band is also filled with high mixing? There actually seems a lack of observations within this band in the observations of Yang et al.

Author's response: The reviewer's concern is reasonable and is worth to be profoundly investigated. In the manuscript, we clarify these limitations of the current model in the last section as follows, "It is noteworthy that despite reasonable agreement between the current simulation and observations, surface forcing, which have potential impact on the modification of ocean stratification and the deep circulation (e.g., Su et al., 2014, 2016a, 2016b; Yang et al., 2015), is not applied to the numerical experiments. Although configured with a buffer zone near the eastern boundary, the experiments are currently configured with closed lateral boundary condition, which cannot simulate the interactions between the processes of the current model domain and the Pacific/Indonesia seas. These limitations may introduce uncertainty to some extent to the simulation results in this study. The potential impact of surface forcing and boundary conditions on the deep circulation in the SCS is worth to be investigated." (lines 299-306)

As for the enhanced mixing configured to the experiments, we followed the results of Yang et al. (2016). We agree that hundreds of profiles and parameterization method are not sufficient to quantitively accurately present the three-dimensional structure of mixing in the SCS. However, the two mixing "hotspots" indicated by the study could be qualitatively robust and consistent with the mechanism of dissipation of internal tides. Therefore, in the current study, instead of applying the exact three-dimensional results of mixing calculated by Yang et al. (2016), we configured the experiments with somehow idealized mixing scheme as far as reflecting the two mixing "hotspots". We clarify the limitations of these configurations to be more specific. (lines 156-161)

Changes in manuscript: We rephrased the sentence and discussed the limitations of no surface forcing and the enhanced mixing in the model configuration of the manuscript.

[#2] Referee Comment: 2) Model spin-up and validation I do not see sufficient validation and assessment of the simulation. A comparison of velocity cross-section with observations (e.g. Fig. 2 only) provides very limited support on the robustness of the simulation results. For instance, how does the model perform in simulating the T/S properties in general for the SCS (at least for the mid- and deep ocean, if not for the whole depth)? Importantly, as the authors deem the dense overflow at the LS to be a crucial factor in determining the dynamics of the WBCs, how is the model behaving in simulating the overflow at the strait? e.g. are the T/S/rho properties of the overflow captured? Are they descending to the bottom basin without being (numerically) diffused? The authors mentioned some numbers on the volume transport of the overflow, but how is the overflow defined?

Besides, regarding the model-data comparison (Fig. 2), "As expected, the control run shows reasonable agreement with the cross-section observations" (L171) seems like an overstatement to me. The WBCs and the return flow are clearly stronger with a broader core compared to the observations. This inconsistency and the possible reasons are not adequately addressed.

As for the model spin-up, the authors need to justify that 20 years are long enough for the model to reach equilibrium or quasi-equilibrium. Evidence should be shown; examples are timeseries of volume transport of the Pacific overflow and the deep WBCs, timeseries of deep basin T/S, and so on.

Author's response: Thanks for this suggestion.

For now, the published and accessible observational data in the deep SCS is relatively sparse. This somehow results to the discrepancy between different climatology atlases, like the WOD01 and GDEM, which are based on different interpolation methods. Both atlases have been used to investigate the deep circulation in the SCS (Fig. RC1, Qu et al., 2006a; Wang et al., 2011). However, the T/S spatial structure of the deep SCS shown in these studies were notably different. Therefore, we did not use these atlases for the model validation. As the key branch of deep circulation, DWBC is the most dominating feature with enhanced velocity and energetic variability. The recent year-long mooring observation on the DWBC in the SCS (Zhou et al., 2017) provides an ideal result for assessing the model simulation. Besides, as the only origin of deep water in the SCS, deepwater overflow through the Luzon Strait have been studied by a series of works, especially on its volume transport. This provides another key result for the assessment of model simulation. Except these two aspects, we do not find other observation results appropriate for validation of model for now. It would be appreciated if the reviewer could provide or suggest where to find more data for the validation.

[Figure]

**Figure RC1. Potential density calculated from the synthetic salinity at 3000 m in the deep South China Sea. The left panel shows $\sigma_2$ based on WOD01 (Qu et al., 2006a) and the right panel shows $\sigma_0$ based on GDEM (Wang et al., 2011).**

In this work, to be consistent and comparable with Zhao et al. (2014), the transport across 121 °E and from the 25th layer (corresponding to $\sigma_2$=36.82 kg m$^{-3}$) to bottom is calculated as the total transport of the deepwater overflow in the Luzon Strait.

The simulated DWBC (4 cm s$^{-1}$) and recirculation are stronger than the observations (2 cm s$^{-1}$) probably due to that the simulated source (1.2 Sv), deepwater overflow in the Luzon Strait, is slightly stronger than the observations (0.88 Sv; Zhou et al., 2014). We noted this in Section 3.1. (lines 183-186)

Fig. RC2 shows the section view of year-mean thickness structure at zonal and meridional section for the control run. The thickness structure was basically stable in the last five years indicated the control run obtained a steady state of the deep circulation in the SCS.

[Figure]

**Figure RC2. Section view of year-mean thickness structure at a zonal section of 16.5 °N (a) and a meridional section of 116 °E (b) for the control run. Thickness numbers and density referenced to 2000 m (σ₂, kg m⁻³) are indicated.**

Changes in manuscript: We added figure and description to clarify this. (lines 163-164)

Minor comments.

[#3] > L9: the first sentence does not need to and should not appear in the abstract;

Author's response: Corrected. (line 9)

[#4]> L35-37: just to clarify, the 'three-dimensional circulation' is one component of the SCS throughflow, and it is the latter that serves as a heat/freshwater conveyor, right?

Author's response: The "South China Sea Throughflow" is another saying of the "three-dimensional circulation" here. We revise the sentence in case of ambiguity. (line 36)

[#5]> L38-52: while the topic of this work is on the SCS deep WBCs, the authors use lots of space introducing the overflow at the LS (I do understand that the WBCs are strongly linked with the overflow). The authors could consider reducing this part.

On the other hand, having given so much background information on the overflow, the authors are expected to describe a bit more on the model representation of overflows, other than merely giving a number of volume transport. See also my earlier comment 2).

Author's response: Following the comment, we reduced the introduction on the deepwater overflow in the manuscript. As for the model representation of overflows, previous study with similar model configurations (Zhao et al., 2014) has investigated the deepwater overflow in the Luzon Strait more detailly. Therefore, we focus on the deep circulation inside the SCS deep basin and may not involve the overflow in detail. (lines 47-53)

[#6]> L41-44: the geographical locations are challenging to the readers. The authors could consider avoiding this, or provide a map showing their locations.

Author's response: We now note the locations of the Bashi Channel, the Luzon Trough and the Taltung Canyon in Fig. 1.

[#7]> L46: rms – full name should be given here.

Author's response: Thanks and corrected as root mean square. (line 47)

[#8]> L68-69: ': : : and higher resolution' – how much higher? e.g. slightly higher than 0.4/0.5 degree, or as high as the resolution in this study?

Author's response: We now note the resolution in She et al. (2014) and Xu and Oey (2014) is 1/12 ° and 1/12 °, respectively. (line 70)

[#9]> L71: what is a/the 'north deep circulation'?

Author's response: We revised the words to be "a stronger deep boundary current along the northern continental slope". (lines 72-73)

[#10]> L76: "Since the DWBC is due to the LS overflow and the beta effect" – this assertion appears to come out of the blue; is this very well known already? Any reference?

Author's response: We rephrase the sentence and add the corresponding references. (lines 78-79)

[#11]> L77: To me the resolution is a secondary consideration. Whether the model produces sufficient overflow water that spills over the sill and whether the water can descend down to the bottom with proper entrainment en route are more important. This could well be my personal biased view though.

Author's response: We agreed that the overflow and the entrainment also play an important role in the deep circulation in the SCS. A short sentence has been added to the text to mention this point. (lines 81-82)

[#12]> L81: I don't see the logical connection between "Due to the lack of field observations" and the rest of the sentence.

Author's response: We rephrased the sentence to avoid being ambiguity. (lines 84-85)

**Deep Circulation in the South China Sea Simulated in a Regional Model**

Xiaolong Zhao[1,2], Chun Zhou[2], Xiaobiao Xu[3], Ruijie Ye[2], Jiwei Tian[2] and Wei Zhao[2]

[1]North China Sea Marine Forecasting Center, State Oceanic Administration, Qingdao, 266061, P. R. China.
[2]Key Laboratory of Physical Oceanography/CIMST, Ocean University of China and Qingdao National Laboratory for Marine Science and Technology, Qingdao 266100, P. R. China.
[3]Center for Ocean-Atmospheric Prediction Studies (COAPS), Florida State University, Tallahassee, FL, USA.

*Correspondence to:* Chun Zhou (chunzhou@ouc.edu.cn)

**Abstract.**  In this study, deep circulation in the South China Sea (SCS) is investigated using results from mesoscale-
[revised manuscript text omitted]
 one of the reasons. Second, different models may have different performances on the entrainment and mixing of ambient water after the deepwater overflow spills into the deep SCS. Third, in most simulations there is a strong cyclonic or anticyclonic circulation cell at the southwest part of the deep circulation under week mixing: a separate cyclonic circulation in Chao et al. (1996) and Shu et al. (2014), while there is an anticyclonic one in Xu and Oey (2014).

Simulation results of the deep circulation in the SCS need to be verified based on observations before being employed to the discussion of the spatio-temporal characteristics of the deep circulation in the SCS.

88 With progresses on the dynamics of submesoscale processes and internal tides (e.g., Su et al., 2018; Yu et al.,

89 2019; Polzin et al., 1996), abyssal enhanced mixing generated by these processes and its impact on the stratification

90 and deep circulation has been drawn increasing attention. Enhanced mixing is a well-observed feature in the SCS.

91 The observations of Tian et al. (2009) and Alford et al. (2011) show diapycnal diffusivity in the SCS and the Luzon

92 Strait increases from about $10^{-3}$ m$^2$ s$^{-1}$ at 1000 m to $10^{-2}$ m$^2$ s$^{-1}$ near the sea floor. This is about two orders of

93 magnitude higher than that in the North Pacific Ocean and is furnished by energetic internal waves induced by the

94 prominent bathymetry in the Luzon Strait (Niwa and Hibiya, 2004; Jan et al., 2007; Tian et al., 2003, 2006). Based

95 on hydrographic measurements with fine scale parameterizations from 335 stations (477 casts), Yang et al. (2016)

96 recently obtained the three-dimensional distribution of turbulent mixing in the SCS for the first time. Two mixing

97 "hotspots" were identified in the bottom waters in the northern shelf of the SCS with the Luzon Strait and the

98 Zhongsha Island Chain areas (their Fig. 4), largely due to internal tide, bottom bathymetry, and near-inertial energy.

99 Previous studies have shown enhanced mixing plays a role in deep circulation in both the Pacific Ocean and the

100 Luzon Strait. Furue and Endoh (2005) indicated the deep Pacific Ocean diffusivity contributes to enhanced

101 production of the Antarctic Bottom Water in the model. The northward transport of the deep meridional overturning

102 circulation across the equator in the Pacific Ocean is stronger with the intense mixing than with weak mixing

103 (Endoh and Hibiya, 2006; their Fig. 3). Zhao et al. (2014) suggested that enhanced mixing in the SCS and the Luzon

104 Strait was the primary driving mechanism for the deep circulation in the Luzon Strait, since it is a key process

105 responsible for the density difference between the Pacific Ocean and the SCS. Based on a simulated tidal mixing

106 scheme, Wang et al. (2017) indicated the tide-induced diapycnal mixing in the Luzon Strait would have a negative

107 effect on driving the cyclonic SCS deep circulation, although without the feature of two mixing "hotspots". Since the

108 mixing is very strong and unevenly distributed in the deep SCS, it is necessary to modify the mixing scheme in the

109 ocean model to be consistent with observed three-dimensional distribution of mixing. Nevertheless, previous

110 numerical studies simulated the deep circulation with homogeneous or simulated vertical mixing parameters in the

111 deep SCS, and one wonders about the sensitivity of the SCS deep circulation to the observed distribution of mixing.

112 Given the lack of observations and inadequate quality control, detailed structures of circulation in the deep SCS

113 have not been mapped out and described adequately. Combining the mooring array in Zhou et al. (2017) with results

114 from mesoscale-eddy-resolving model simulations, the present study for the first time investigates deep circulation

115 under enhanced mixing two mixing "hotspots" in the SCS. The paper is organized as follows. After the introduction, the data and model configuration are described in Sect. 2. Section 3.1 presents the model results compared with observations. Section 3.2 is devoted to the horizontal pattern of mean circulation. Variability of deep circulation is discussed in Sect. 3.3, and Sect. 3.4 examines sensitivity to distribution of mixing. A summary and discussion follows in Sect. 4.

**2. Data and Model Configuration**

As part of the SCS mooring array, an array of six bottom-anchored moorings was deployed off the eastern slope of the Zhongsha Islands between 28 August 2012 and 11 January 2014 (M1-M6, see Fig. 1 for locations). Twenty-nine

Aanderaa Data Instruments RCM Seaguard current meters were utilized to measure the horizontal current of the

DWBC at nominal depths of 2000 m, 2500 m, 3000 m, 3500 m, and 4000 m, with generally 500 m resolution vertically. Details pertinent to these moorings are shown in Table 1. All current meters were configured to record data at a sample interval of one hour. Detailed results are discussed in Zhou et al. (2017). Here, we use the observed mean velocity section to examine the simulated time mean structure of the DWBC.

The regional simulation is similar to that of Zhao et al. (2014). The general circulation model used was the Hybrid

Coordinate Ocean Model (HYCOM; Bleck, 2002; Chassignet et al., 2003) configured with a horizontal resolution of

$1/12°$ (~9 km resolution in our area of interest). The computational domain, which extends from 4 °N to 25 °N and

105 °E to 125 °E (Fig. 1), includes the SCS and part of the northwestern Pacific Ocean. A total of 32 vertical hybrid layers are configured with density referenced to 2000 m ($\sigma_2$, kg m$^{-3}$): 28.10, 28.90, 29.70, 30.50, 30.95, 31.50, 32.05,

32.60, 33.15, 33.70, 34.25, 34.75, 35.15, 35.50, 35.80, 36.04, 36.20, 36.34, 36.46, 36.56, 36.64, 36.70, 36.74, 36.78,

36.82, 36.84, 36.86, 36.88, 36.92, 36.96, 37.01, and 37.06. The bottom topography is from version 13.1 of Smith and Sandwell (1997) with  1/60 ° resolution. The simulation was initialized with rest and January temperature and salinity fields from the third version of monthly 1/4 ° ocean climatology GDEM (Carnes, 2009).

Since the current work is designed to be a process study, surface forcing was not applied in the experiments. All lateral boundaries were closed with no normal flow, within a 19-grid buffer zone near the eastern boundary, the modeled temperature and salinity are restored toward the same (monthly)

climatology with an e-folding time of 0.5-32 days that increased with distance from the boundary. The bottom stress was parameterized using a quadratic drag law at the lowest 10 m, with a constant drag coefficient $C_D = 2.5 \times 10^{-3}$.

Based on similar configurations with all of the numerical experiments started from rest and integrated for 10 years,

Zhao et al. (2014) studied the deep water circulation in the Luzon Strait, which was in good agreement with the observations based on repeated conductivity-temperature-depth (CTD) and lowered acoustic Doppler current profiler (LADCP) surveys. We modified the K-profile parameterization (KPP; Large et al., 1994) mixing scheme in accordance with the two observed mixing "hotspots" found in Yang et al. (2016). Thus, the control run was configured with larger vertical mixing parameters, in which the diapycnal diffusivity beneath 1000 m were set to $10^{-3}$

$\mathrm{m^2\ s^{-1}}$ in both the north shelf of the SCS with the Luzon Strait (109-122 °E, 18-23 °N) and the Zhongsha Island

Chain area (109-122 °E, 14-17 °N, red boxes in Fig. 1). To examine the impact of mixing, four sensitivity experiments were used with the same configuration as the control run, but with different mixing schemes: Following

Zhao et al. (2014), Exp-5 and Exp-3 were configured with the native KPP scheme as background mixing of $10^{-5}\ \mathrm{m^2}$

$\mathrm{s^{-1}}$ and the diapycnal diffusivity beneath 1000 m in the SCS and the Luzon Strait (west of 122 °E) as $10^{-3}\ \mathrm{m^2\ s^{-1}}$, respectively. Exp-3A and Exp-3C were configured with the lager vertical mixing parameters in different areas, in which the diapycnal diffusivity beneath 1000 m were set to $10^{-3}\ \mathrm{m^2\ s^{-1}}$ in the north shelf of the SCS with the Luzon

Strait (109-122 °E, 18-23 °N) and the Zhongsha Island Chain area (109-122 °E, 14-17 °N), respectively. Instead of applying the exact results of mixing distribution of Yang et al. (2016), these configurations are idealized to some extent, in order to reproduce the two mixing "hotspots" dynamically explained by dissipation of internal tides, while not following the specific distribution and magnitude which still need to be verified due to the limitations of numbers of CTD profiles and parameterization method. These configurations may somehow introduce uncertainty to the simulation results which is difficult to evaluate with the current observations.

In order to obtain a steady state of the deep circulation in the SCS, we integrated all of the numerical experiments for 20 years and averaged the last five years as the simulated annual mean results mentioned below (as shown in Fig.

2, the thickness structure was basically stable in the last five years indicated the control run has been stable during the last 10 years).

**3. Key Results**

[revised manuscript text omitted]

2015), is not applied to the numerical experiments. Although configured with a buffer zone near the eastern boundary, the experiments are currently configured with closed lateral boundary condition, which cannot simulate the interactions between the processes of the current model domain and the Pacific/Indonesia seas. These limitations may introduce uncertainty to some extent to the simulation results in this study. The potential impact of surface forcing and boundary conditions on the deep circulation in the SCS is worth to be investigated.

**Data availability**

Model outputs are available upon request to the first author.

**Author contribution**

[revised manuscript text omitted]